# Unified single-cell analysis of testis gene regulation and pathology in five mouse strains

Min Jung[1†], Daniel Wells[2,3†], Jannette Rusch[1], Suhaira Ahmad[1], Jonathan Marchini[2,3], Simon R Myers[2,3*], Donald F Conrad[1,4*]

[1]Department of Genetics, Washington University School of Medicine, St. Louis, United States; [2]The Wellcome Centre for Human Genetics, University of Oxford, Oxford, United Kingdom; [3]Department of Statistics, University of Oxford, Oxford, United Kingdom; [4]Division of Genetics, Oregon National Primate Research Center, Oregon Health & Science University, Portland, United States

**Abstract** To fully exploit the potential of single-cell functional genomics in the study of development and disease, robust methods are needed to simplify the analysis of data across samples, time-points and individuals. Here we introduce a model-based factor analysis method, SDA, to analyze a novel 57,600 cell dataset from the testes of wild-type mice and mice with gonadal defects due to disruption of the genes *Mlh3*, *Hormad1*, *Cul4a* or *Cnp*. By jointly analyzing mutant and wild-type cells we decomposed our data into 46 components that identify novel meiotic gene-regulatory programs, mutant-specific pathological processes, and technical effects, and provide a framework for imputation. We identify, de novo, DNA sequence motifs associated with individual components that define temporally varying modes of gene expression control. Analysis of SDA components also led us to identify a rare population of macrophages within the seminiferous tubules of *Mlh3*[-/-] and *Hormad1*[-/-] mice, an area typically associated with immune privilege.
DOI: https://doi.org/10.7554/eLife.43966.001

**\*For correspondence:**
myers@stats.ox.ac.uk (SRM);
conradon@ohsu.edu (DFC)

[†]These authors contributed equally to this work

**Competing interests:** The authors declare that no competing interests exist.

## Introduction

The testis is an amalgamation of somatic cells and germ cells that coordinate a complex set of cellular interactions within the gonad, and between the gonad and the rest of the organism (*Figure 1A*). The key function of the testis is to execute spermatogenesis, a developmental process that operates continually in all male adult mammals in order to generate genetically diverse gametes through recombination and independent assortment of homologous chromosomes. The mechanisms of this process are important for the evolution, fertility and speciation of all sexually reproducing organisms.

A deeper understanding of the transcriptional program of spermatogenesis has potential applications in contraception (*Schultz et al., 2003*), in vitro sperm production for research and the treatment of infertility (*Zhou et al., 2016*), and the diagnosis of infertility, among others. Prior to the advent of single-cell genomics, studies of the highly dynamic transcriptional programs underlying sperm production were limited by the cellular complexity of the testis - comprised of at least seven somatic cell types, and at least 26 morphologically distinct germ cell classes (*Hess and de Franca, 2009*).

The testis has a number of unique features: its transcriptome has by far the largest number of tissue specific genes (over twice as many as the 2nd ranked tissue the cerebral cortex – with which the testis shares an unusual similarity) (*Djureinovic et al., 2014*; *Guo et al., 2005*; *Uhlén et al., 2015*); it

**Figure 1.** Mapping cellular diversity in the adult testis using single-cell expression profiling. (**A**) Anatomy of the testis. Adult testis are comprised of germ cells (spermatogonia, primary spermatocytes, secondary spermatocytes, spermatids and spermatozoa) and somatic cells. Within the seminiferous tubules, there is a population of somatic cells (Sertoli). The tubules are wrapped by muscle-like 'myoid' cells. Outside the tubules are a heterogeneous, poorly defined population of 'interstitial' somatic cells including Leydig cells and telocytes. (**B**) Overview of the experiments. To establish the utility of single-cell profiling for testis phenotyping, we performed a series of experiments (i) comparing the quality of traditional enzymatic dissociation and more rapid mechanical dissociation, (ii) comparing the expression profiles of cells from total testis dissociation to testicular cells of known identity purified by FACS, (iii) comparing expression profiles of wild-type animals to cells isolated from four mutant strains with testis phenotypes (*Figure 1—figure supplement 1*). (**C**) We used Drop-seq to profile 26,200 cells from wild-type animals and 31,400 cells from mutant animals, with an average of 1155 transcripts/cell and 725 genes/cell (wild-type) and 2223 transcripts/cell and 1133 genes/cell (mutants). (**D**) We applied SDA and used t-SNE to visualize cells colored by k-means clustering of 20,322 cells, derived from our full dataset of wild-type and mutant animals, into 32 clusters (Materials and methods, *Figure 1—figure supplements 1–5*). Label assignment clearly indicates a spatial organization of testis cells in t-SNE space, with somatic cell populations flanking the germ cells in small pockets. The full set of 32 clusters has been simplified into 12 major classes for ease of interpretation; the full clustering is shown in *Figure 1—figure supplement 2*. (**E**) Histology sections from wild-type and mutant testis, illustrating the phenotypes observed in wild-type and the four mutant strains characterized by Drop-seq. Three of the strains, $Mlh3^{-/-}$, $Hormad1^{-/-}$ and $Cul4^{-/-}$ have known pathology, while strain *CNP* represents an unpublished transgenic line with spontaneous male infertility. (**F**) Mapping of cells from each mouse strain into t-SNE space (colored points) compared to the background of all other strains (gray points). Mutant strains occupy distinct locations within t-SNE space, reflecting the absence of certain cell types in some strains (e.g. $Mlh3^{-/-}$ and $Hormad1^{-/-}$), and alteration of expression in remaining cells (e.g. $Hormad1^{-/-}$). (**G**) Counting individual cell types provides a quantitative phenotype of cellular heterogeneity in each strain.
DOI: https://doi.org/10.7554/eLife.43966.002

The following figure supplements are available for figure 1:

**Figure supplement 1.** Comparison of effects of dissociation protocols and mutation status on cell ascertainment and single-cell gene expression.
DOI: https://doi.org/10.7554/eLife.43966.003
**Figure supplement 2.** Mapping the Cellular Diversity of the Testis.
DOI: https://doi.org/10.7554/eLife.43966.004
**Figure supplement 3.** Overview of expression patterns for some well known testis cell markers in t-SNE space.
DOI: https://doi.org/10.7554/eLife.43966.005
**Figure supplement 4.** Tabulation of cluster counts by mouse strain and differential expression analysis within clusters.
DOI: https://doi.org/10.7554/eLife.43966.006
**Figure supplement 5.** Dissection of Somatic Cell Population Heterogeneity.
DOI: https://doi.org/10.7554/eLife.43966.007

contains the only cells in the male body with sex chromosome inactivation (*Yan and McCarrey, 2009*); meiotic cells undergo programed double strand break formation, homologous chromosome pairing, and recombination; cells undergoing meiosis share transcripts through cytoplasmic bridges (*Braun et al., 1989*); and it features dramatic chromatin remodeling, when the majority of histones are stripped away during spermiogenesis and replaced with small, highly basic proteins known as protamines (*Hammoud et al., 2009*).

Use of genetic tools has enabled the dissection of the homeostatic mechanisms that regulate spermatogenesis, revealing both cell autonomous and non-autonomous mechanisms. However, most perturbations that disrupt spermatogenesis also change the cellular composition of the testis, frustrating the use of high throughput genomic technologies in the study of gonadal defects. By removing heterogeneity as a confounding factor, single cell RNA sequencing (scRNA-seq) promises to revolutionize the study of testis biology. Likewise, it will completely change the way that human testis defects are diagnosed clinically, where testis biopsy is the standard of care for severe cases of male infertility (*Dohle et al., 2012*).

Here, we performed scRNA-seq on 57,600 cells from the mouse testis, using wild-type animals and four mutant lines with defects in sperm production (*Figure 1B*). We set out to develop an analysis approach that would allow us to extract mechanistic insights from joint interrogation of these multiple mouse strains; to gain insights into spermatogenesis and its regulation, using the precise resolution of single-cell analysis; and to establish the utility of scRNA-sequencing for dissecting testis gene regulation in both normal and pathological states.

To do this, we leverage a recently developed factor analysis method, called sparse decomposition of arrays (SDA), which has not previously been applied to single-cell RNA-seq data, and demonstrate how it can be used on scRNA-seq data for cleanup and imputation, identification of co-regulated genes, and to create a dictionary of disease from a joint analysis of mutant and wild-type animals. We show that, unlike standard clustering, we are able to decompose expression patterns into

temporally overlapping yet distinct components, which each possesses specific regulatory mechanisms and functions, providing new insights relative to recent reports of scRNA-seq from mouse testis (*Chen et al., 2018*; *Ernst et al., 2019*; *Green et al., 2018*; *Hermann et al., 2018*; *Lukassen et al., 2018*). Moreover, we retain the ability of other existing scRNA-seq analysis methods to order cells from early to late meiosis, and to identify distinct groups of non-meiotic cells.

## Results

### Mapping the cellular diversity of the testis with single-cell RNA-seq

To isolate individual cells for data generation, we initially tested two methods for testis dissociation: enzymatic dissociation, a slow 2 hour protocol, vs. a rapid 30 minute protocol based on mechanical disruption (*Lima et al., 2017*). Single cell expression profiles from the two methods showed excellent agreement (r = 0.95), with no important differences in cell quality or ascertainment (*Figure 1*, *Figure 1—figure supplement 1*), so we applied the mechanical dissociation approach for further experiments (*Supplementary file 1*). We performed scRNA-seq to generate 25,423 cell profiles isolated from total testis dissociations of 11 wild-type animals (WT1-WT11). We compared these to reference data for 296 spermatogonia, 199 primary spermatocytes, 398 secondary spermatocytes, and 299 spermatids, purified by FACS (Materials and methods). Transcript yield per wild-type cell (*Figure 1C*, *Supplementary file 2*) were consistent with previous studies using DropSeq on testicular cells (*Green et al., 2018*) or different cell types.

We added to this an additional 31,400 single cell profiles from four different mutant mouse strains exhibiting spermatogenesis defects: three mutants with known molecular mechanisms (knockouts of *Mlh3*, *Hormad1*, and *Cul4a*) as well as one knockin of a transgene (*Cnp*) that led to idiopathic infertility. We performed histological confirmation of testis defects in each animal prior to sequencing (*Figure 1E*). Seminiferous tubules in *Mlh3*[-/-] and *Hormad1*[-/-] mice exhibited complete early meiotic arrest and absence of spermatozoa. *Cul4a*[-/-] sections showed partial impairment of spermatogenesis, with a significant decrease in number of post-meiotic cells and abnormal spermatids. Sections from both *Cul4a*[-/-] and *Cnp* mice presented giant multinucleated cells, but this type of cell was much more prevalent in *Cnp* seminiferous tubules. *Cnp* mice displayed a clear defect in spermatogenesis; the number of elongating spermatids was grossly reduced to compared to wild-type, and the few elongating spermatids seen in the histology sections featured misshapen nuclear morphology and odd orientation within the disorganized tubules. Sperm tails were occasionally seen in the lumen. Further molecular analysis is required to firmly characterize which stage(s) of spermatogenesis are affected.

### Application of SDA, and comparison to classical clustering analysis

One specific challenge of analyzing a developmental system is that cluster-based cell type classification might artificially define hard thresholds in a more continuous process. Furthermore, a single cell's transcriptome is a mixture of multiple transcriptional programs, some of which may be shared across cell types. In order to identify these underlying transcriptional programs themselves rather than discrete cell types we applied SDA (*Hore et al., 2016*). This is a model-based factor analysis method to decompose a (cell by gene expression) matrix into sparse, latent factors, or 'components' that identify co-varying sets of genes which, for example, could arise due to transcription factor binding or batch effects (Materials and methods). Each component is composed of two vectors of scores: one reflecting which genes are active in that component, and the other reflecting the relative activity of the component in each cell, which can vary continuously across cells, negating the need for clustering. This framework provides a unified approach to simultaneously soft cluster cells, identify co-expressed marker genes, and impute noisy gene expression (Materials and methods). We inferred 50 components using SDA. Using these components, we visualized the overall results using t-distributed Stochastic Neighborhood Embedding (t-SNE) (Materials and methods, *Figure 1D*): this t-SNE projection is also used in many subsequent analyses. We estimated the developmental ordering of cells using pseudotime modeling (Materials and methods), based on our t-SNE embedding.

First, to provide a cross-check for our SDA results, we performed k-means (hard) clustering of our single cell libraries into discrete groups. (Materials and methods, *Supplementary file 3*, *Supplementary file 4*). We visualized the resulting 32 distinct clusters in t-SNE space

(Materials and methods, *Figure 1D*, *Figure 1—figure supplement 2*). Next, by inspecting the expression levels of known cell type markers and comparing to the FACS-sorted cells, we could resolve our 32 clusters into 11 distinct subtypes of germ cells and four somatic cell populations – Leydig cells, Sertoli cells, immune cells, and telocytes (*Figure 1—figure supplement 2* and *Figure 1—figure supplement 3*). By tallying counts of cells within each cluster, we generated a digital readout of the cellular composition of wild-type and mutant animals (*Figure 1G*, *Figure 1—figure supplement 4*), and are able to associate each SDA component to expression activity in particular cell type(s).

Careful examination and quantification of cell-type composition differences in each mutant strain recapitulated the known pathology of mutants (*Mlh3*[-/-], *Hormad1*[-/-] and *Cul4a*[-/-]) at digital resolution. The location of mutant cells in t-SNE space illustrated the absence of certain cell types within spermatogenesis (*Figure 1F*). Consistent with the known biology, we observed that both *Mlh3*[-/-] and *Hormad1*[-/-] cells arrest at different stages of meiosis I; mid-pachytene and leptotene/zygotene respectively. Derangement of certain cell types in the developmental trajectory was also observed as leptotene/zygotene *Hormad1*[-/-] cells formed distinct clusters. Both t-SNE and hard clusters indicated strong mixing of mutant and wild-type cells; of the 32 clusters, only two did not contain both wild-type and mutant cells. Both lacked wild-type cells: cluster 9, a Sertoli cell cluster, and cluster 30, containing leptotene spermatocytes primarily from *Hormad1*[-/-]. As the bulk of our experiments were performed on total testis samples, we do see preferential ascertainment of some cell types from the mutant strains depleted of post-meiotic germ cells: 95% of somatic cells (clusters 1–5,8,9) and 83% of pre-pachytene germ cells (clusters 6, 30–32) are derived from mutants (*Figure 1—figure supplement 4A*). The majority of these clusters have fewer than 10 genes with differential expression detectable between mutant and wild-type (*Figure 1—figure supplement 4B*), and we proceeded with a joint analysis of mutant and wild-type cells, with the caveat that conclusions about the biology of these particular clusters are derived largely from mutant strains.

Following these comparisons, we interpreted our SDA results using published bulk RNA-seq testis data, histology, and GO category analyses.

## New molecular markers of cellular subtypes

Single cell RNA sequencing provides new opportunities to assess important open questions in the field of spermatogenesis. Along with the expected patterns of expression for known markers, we identified numerous novel markers for all populations, some of which we selected for validation using immunohistochemistry (*Figure 2*). Noteworthy is the identification of KIF5B as a Sertoli cell protein that provides more extensive coverage of the cell body than the conventional markers TUBB and VIM, and the identification of ABHD5 as a marker for the subcellular structure of developing germ cells known as the residual body. Protein products for predicted markers ACYP1, UNC80, and CCDC62 were not detected, which might be an antibody-related problem or an indication that these RNAs were not translated.

We identified a number of somatic cell populations (hard clusters 1,2,3,4,5, 8 and 9 in *Figure 1—figure supplement 2A* 'Merged'). Because our SDA analysis suggested multiple components, varying even within these clusters (see below), we performed additional targeted hard clustering analyses on these cells (Materials and methods), identifying additional complexity: 10 identified somatic cell clusters comprise 4 Sertoli cell sub-clusters, 3 Leydig sub-clusters, two immune cell clusters (macrophages and lymphocytes) and one telocyte cluster (*Figure 1—figure supplement 5*). Telocytes are a recently reported stromal cell type present in a wide range of tissues, and are little studied in testis (*Marini et al., 2018*). In addition to the previously reported markers *Cd34* and *Pdgfra*, we find a number of even more highly specific expression markers for telocytes, including *Dcn*, *Gsn*, *Tcf21* (*Supplementary file 2*).

Each Sertoli sub-cluster is enriched with differing GO terms (biological processes) including cytoskeleton organization (sub-cluster 1), protein folding (sub-cluster 2), RNA splicing (sub-cluster 2 and 3) and spermatogenesis (sub-cluster 4), while Leydig sub-clusters are enriched for steroid and lipid biosynthetic process (sub-cluster 1), ATP synthesis coupled electron transport and drug metabolic process (sub-cluster 2) and cofactor and steroid metabolic process (sub-cluster 3) (*Figure 1—figure supplement 5D*).

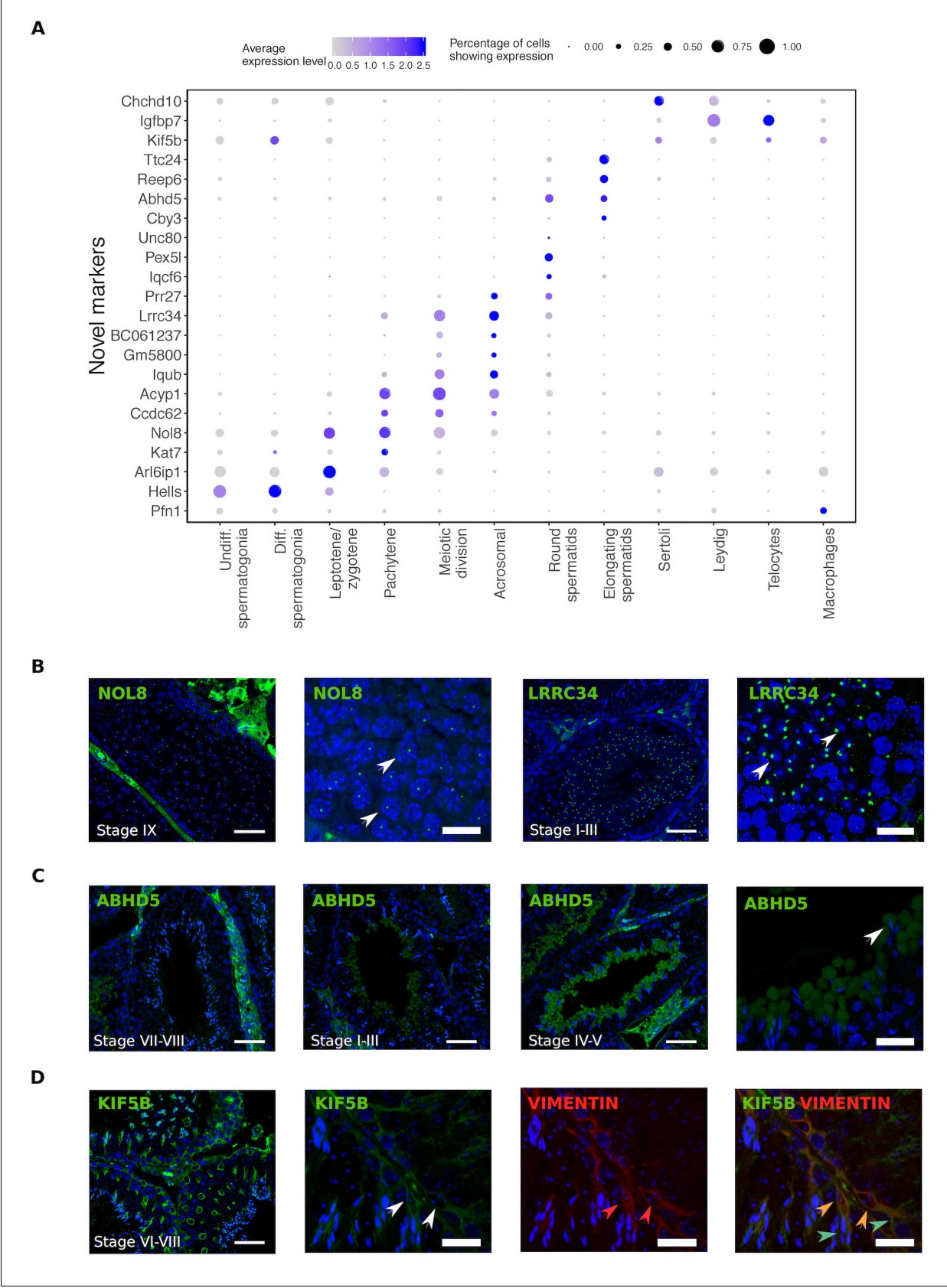

**Figure 2.** Identification of novel cellular markers from single-cell data. (A) Across major cell-type clusters, we identified 22 gene expression markers specific to one cell type or aspect of spermatogenesis and not previously reported. Here we show the expression levels of these genes. Expected protein expression patterns for *Nol8*, *Lrrc34*, *Abhd5*, and *Kif5b* were confirmed, but the antibodies for *Acyp1*, *Ccdc62*, and *Unc80* did not show positive staining in any testicular cell types, which could be an antibody-related problem or an indication that these RNAs were not translated. (B–D) Thin scale bar, 50 μm; thick scale bar, 20 μm. (B) *Nol8*, a nucleolar protein, marks primary spermatocytes while *Lrrc34* marks nucleoli in round spermatids (white arrowheads) (C) Within the tubules, *Abhd5* marks specific cytoplasmic regions of elongating spermatids destined to form the residual body (white arrow head) and staining intensity peaks during seminiferous tubule stages IV-V. (D) *Kif5b* marks Sertoli cells within seminiferous tubules (white arrow head). We co-stained *Kif5b* with a well-known Sertoli cell marker, Vimentin (red arrow head), and indeed both proteins colocalize to Sertoli cells (orange arrow head). Co-staining also reveals that *Kif5b* staining extends further out in the cell body (blue-green arrow head) than Vimentin.
DOI: https://doi.org/10.7554/eLife.43966.008

## SDA-based gene expression modules

Based on the above analyses, it is clear that our 50 SDA components represent all the major different cell types and developmental stages of spermatogenesis, with other specific components capturing batch effects and general processes such as respiration. Encouragingly, most components contained relatively few highly expressed genes (*Figure 3B*) and, when compared to alternative commonly used methods for matrix factorisation (non-negative matrix factorization, NNMF; principal component analysis, PCA; independent component analysis, ICA), SDA produced the most sparse model (*Figure 3C*). Although the model used for inference is symmetric for positive/negative gene weightings, many identified components showed strong biases towards positive or negative weights, consistent with expectations for identifying a group of co-activated (or co-repressed) genes (e.g. *Figure 3E*). Likewise, the cell loadings of each component frequently highlight specific cellular subsets that localize in t-SNE space and pseudotime (*Figure 3D&F*, *Figure 3—figure supplement 1*), and often interpretable as particular identifiable cell types in our initial hard clustering. Thus, we label SDA components as 'expression modules'. We found that most components generated from an SDA analysis of only wild-type data were also observed in the joint analysis of wild-type and mutant data, which we proceed to use for the remaining analyses (*Figure 3—figure supplements 2–4*).

To provide further intuition towards how SDA components summarize transcriptional programs, we selected 14 components that, collectively, load highly on germ cells throughout spermatogenesis. When we visualize the total expression output for each cell, ordered by pseudotime, as a sum of all 14 components, it is clear that expression can be modeled as an overlapping series of components in time, coming on and going off gradually over different timescales (*Figure 4A–B*). Each component is enriched for specific genes, and, importantly, genes with different identified biological functions (*Figure 4C&D*). SDA components provide complementary information to hard clustering: a single hard cluster may have significant cell scores from as many as three components, indicating multiple different expression programs jointly active in each cell. Conversely a single SDA component may show significant cell scores across more than three hard clusters (*Supplementary file 3*), emphasizing that expression changes gradually as cell types and fates evolve (*Figure 4—figure supplements 1–2*).

In addition to identifying soft clusters and their markers, by multiplying the cell scores and gene loadings, SDA can impute very sparse, noisy, expression data. In principle, harnessing the correlation structure of gene coexpression across cells can improve predictions, overcoming the sparsity of the initial data. Indeed, our dataset has 93.8% zero values and a median of 1,312 UMI transcripts per cell. Nonetheless, SDA imputation is able to estimate expression of individual genes even when in many cells zero reads are observed (*Figure 5A*). It is not possible to determine the true expression vector for an individual cell, so we use cross-validation to test whether imputation improves expression estimates. Specifically, we assign each read to either a training or test set. We predict gene expression based on the training set, using the SDA approach, or another approach (e.g. the dedicated single cell imputation method MAGIC; *van Dijk et al., 2018*), and then evaluate our ability to rank gene expression using the test set (Materials and methods). SDA imputation outperforms approaches using the raw data, for essentially all cells in the test data (*Figure 5B&C*, *Figure 5—figure supplement 1C*). While providing the most sparse representation (*Figure 3C*), SDA still imputes equally well, compared to other matrix factorizations and to MAGIC (*van Dijk et al., 2018*) (*Figure 5C*, *Figure 5—figure supplement 1A*). Further, when compared to NNMF, SDA provides

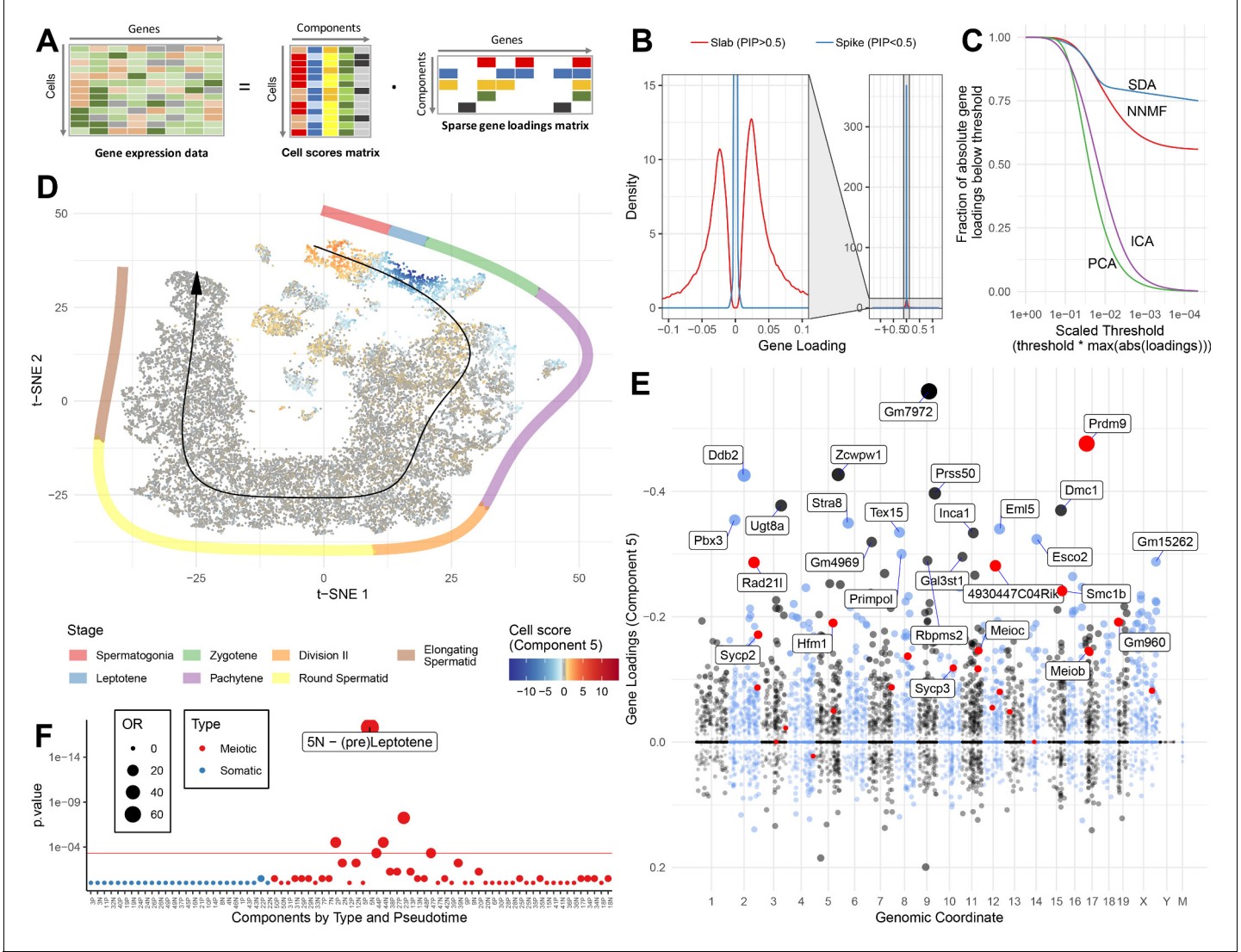

**Figure 3.** SDA identifies gene modules and maps them to cells. (**A**) We applied sparse decomposition analysis (SDA) to identify latent factors ('components') representing gene modules. These components are defined by two vectors – one that indicates the loading of each cell on the component, and one that indicates the loading of each gene on the component. (**B**), SDA uses a spike and slab prior on the gene loadings to induce sparsity (a point mass at 0 and a centered normal distribution respectively). PIP = Posterior Inclusion Probability that a gene loading is not equal to zero (i.e. not in the spike). The figure shows the density of gene loadings over all components with loadings separated into genes with PIPs > 0.5 (20%) versus <0.5, indicating the sparsity of resulting gene loadings. (**C**) SDA produces sparser representation of gene loadings compared to other matrix factorizations: NNMF, ICA and PCA. For each method, the fraction of all absolute gene loadings exceeding a 'no loading' sparsity threshold is shown, normalized by the maximum absolute loading across all components for that method. (**D**) We fitted 50 SDA components using 20,322 wild-type and KO cells (see also *Figure 3—figure supplements 1–5*). We illustrate component 5. The loadings of component 5 in t-SNE space highlight a cluster of cells at the leptotene early meiotic developmental stage. Black arrow: the principle curve fit to the germ cell data, corresponding to the developmental ordering of each cell progressing through spermatogenesis. The colored segmented line shows broad staging of spermatogenesis. (**E**) Genomic location versus loadings for component 5. Most genes have near-zero loadings, but a fraction have non-zero loadings, including the well-known histone methyltransferase *Prdm9*. Red genes: GWAS hits for human recombination rate. (**F**) Component 5 is highly and specifically enriched for GWAS hits of human recombination rate. OR: Odds Ratio. P value by FET (main text). Positive (**P**) and negative (**N**) loadings are tested separately. For one-sided components (cell score range ratio >5) the minor side is omitted. Red horizontal line: p=0.05 after Bonferroni correction for multiple testing.

DOI: https://doi.org/10.7554/eLife.43966.009

The following figure supplements are available for figure 3:

**Figure supplement 1.** Overview of cell score loadings in t-SNE space for all components produced by SDA except single cell components (1, 4, 8, 14, and 46).

DOI: https://doi.org/10.7554/eLife.43966.010

*Figure 3 continued on next page*

*Figure 3 continued*

**Figure supplement 2.** Robustness of SDA Results.
DOI: https://doi.org/10.7554/eLife.43966.011
**Figure supplement 3.** Rotation Matrix.
DOI: https://doi.org/10.7554/eLife.43966.012
**Figure supplement 4.** Correlation of C31 gene loadings.
DOI: https://doi.org/10.7554/eLife.43966.013
**Figure supplement 5.** Robustness of t-SNE embedding.
DOI: https://doi.org/10.7554/eLife.43966.014

additional biological insights for the same number of components (Materials and methods; *Figure 5D,E,F* & *Figure 5—figure supplement 1B*). In addition to obviating the need for further clustering and differential expression analyses, an advantage of using matrix factorization for imputation is the much smaller memory footprint required to store the results: on our dataset MAGIC data is 2.9 Gb whereas the SDA matrices are just 18 Mb (12.6 Mb when loadings with PIP <0.5 are set to 0).

Overall, of 50 components, six represent batch effects, five are components with only a single cell, 13 are observed only in somatic cell types, 23 only in germ cells, and three components load on both somatic and germ cells (*Figure 3—figure supplement 1*). Within somatic cell components, we observe components corresponding to Sertoli cells (n = 4), Leydig cells (4), macrophages (1), T lymphocytes (1), telocytes (1), peritubular myoid cells (1) as well as an interesting component that seems expressed in all interstitial cells (1). Among germ cell-specific components, we observe components corresponding to processes active in spermatogonia (5), preleptotene spermatocytes (1), leptotene/zygotene (2), pachytene (5), diplotene (1), and spermiogenesis (7). Thus, we find multiple sub-components within existing recognized meiotic stages, adding considerable resolution relative to bulk-sequencing approaches. For some analyses below, we considered positively and negatively weighted genes within a component separately, in case these represent different modes of regulation, within the same groups of cells. We provide a web application to enable interactive exploration of gene expression and components at http://www.stats.ox.ac.uk/~myers/testisAtlas.html.

Prior to single cell studies such as our own, previous approaches to germ cell transcriptional profiling provided a single, static summary of pachytene expression from bulk sequencing of purified cells (*da Cruz et al., 2016*; *Soumillon et al., 2013*). Here, we are able to decompose pachytene gene regulation into five components (13, 39, 42, 47, and 48). Although component cell loadings overlap in pseudotime, they differ dramatically in their dynamics (*Figure 4A&B*). For instance, component 13 and 47 loadings fluctuate between positive and negative, while component 42 loading is constantly negative when active (*Figure 3—figure supplement 1*). The genes with strong loadings within expression components do not necessarily associate with a single, coherent functional process, nor even a set of co-translated transcripts. Instead, components 13, 39, 42 and 48 each involve both a substantial number of genes required for meiosis, but also genes required for postmeiotic processes, including sperm tail formation (*Supplementary file 3*).

## Components reflect known biology but also highlight sets of genes with mysterious purpose

Five components correspond to processes in spermatogonia. Component 31 represents undifferentiated spermatogonia expressing *Zbtb16* (aka *Plzf*) (*Buaas et al., 2004*) and *Foxo1* (*Goertz et al., 2011*), while component 50 splits these into two subpopulations one expressing, *Gfra1* (*He et al., 2007*) and *Glis3* (*Kang et al., 2016*), and the other *Nanos3* (*Suzuki et al., 2009*), *Lin28a* (*Zheng et al., 2009*) and *Foxf1* (*Figure 5D*). Component seven likely represents $A_{1-4}$ spermatogonia expressing *Glis2*, *Nanos1*, *Kit*, and *Stra8*. Component two includes *Ctcfl*, *Pou4f1*, and *Esx1* - likely representing intermediate and type B spermatogonia and component 33 is a broader spermatogonial component enriched in genes involved in spermatogonial differentiation (*Supplementary file 3*).

During meiosis an extended prophase I (lasting 14 days in mice) is itself divided into stages: Leptotene, Zygotene, Pachytene, and Diplotene (*Oakberg, 1956*). During prophase homologous chromosomes pair to enable genetic recombination and balanced segregation during meiotic divisions. In the earlier stages homologous chromosomes begin to associate aided by meiosis-specific cohesin

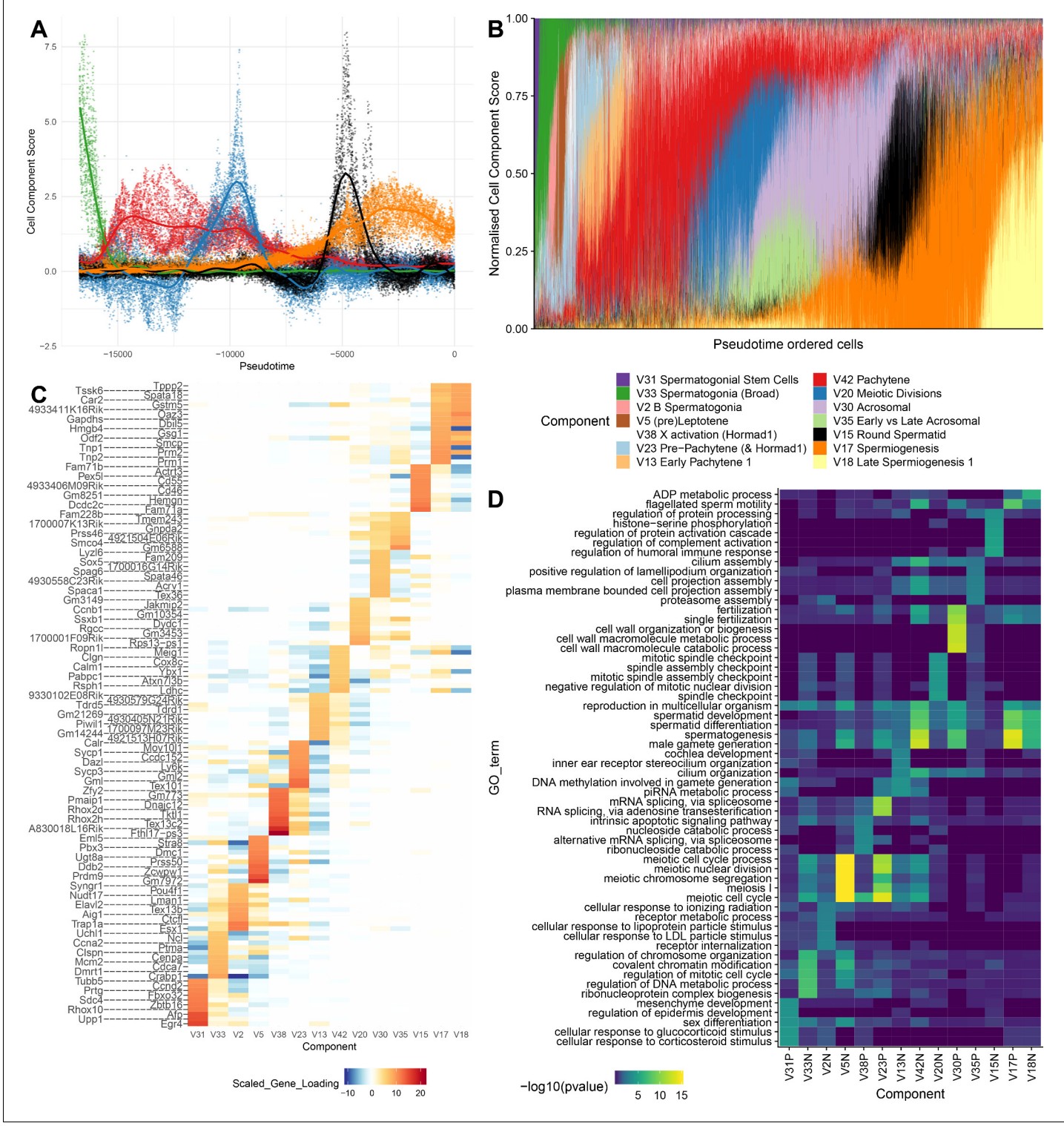

**Figure 4.** SDA components overlap but represent distinct processes. (**A**) For five example components, the cell scores for each cell are plotted through pseudotime, indicating strongly overlapping dynamically varying component activity. Component signs were chosen to be mainly positive (components have arbitrary sign). Color mappings as in panel B. (**B**) Stacked bar plot of cell component loadings for 14 germ components sorted by cell pseudotime. Each column corresponds to an individual cell and the total positive component loadings for each are normalized to one after flipping components to be mainly positive. Factorization by SDA indicates that transcription during spermatogenesis can be represented as an overlapping series of components in time, coming on and off gradually on different timescales. See also *Figure 4—figure supplements 1–2* for alternative visualizations of components in pseudotime. (**C**) Furthermore, these components are comprised of distinct gene sets driving distinct biological processes. Shown are

*Figure 4 continued on next page*

*Figure 4 continued*

the top 10 gene loadings for each of the components in (B) represented as a heatmap. Most genes have strong loading on only one component. (D) Likewise, a gene ontology enrichment analysis for biological processes in the top 250 genes for each component indicates largely non-overlapping enrichments across components. More in-depth analysis of GO enrichments and gene loadings for each component allow separation of components into biological and technical effects (*Figure 4—figure supplements 3–4*).

DOI: https://doi.org/10.7554/eLife.43966.015

The following figure supplements are available for figure 4:

**Figure supplement 1.** Heatmap of SDA component scores.

DOI: https://doi.org/10.7554/eLife.43966.016

**Figure supplement 2.** Overview of Individual SDA Components.

DOI: https://doi.org/10.7554/eLife.43966.017

**Figure supplement 3.** Detailed Analysis of Individual SDA Components.

DOI: https://doi.org/10.7554/eLife.43966.018

**Figure supplement 4.** Components representing batch effects and cellular respiration.

DOI: https://doi.org/10.7554/eLife.43966.019

and telomeric tethering to the nuclear envelope (*Boateng et al., 2013*; *Ishiguro et al., 2014*). Several hundred *Spo11*-induced programed double-strand breaks (DSBs) then occour at *Prdm9*-marked sites (*Baudat et al., 2010*; *Keeney et al., 1997*; *Myers et al., 2010*; *Parvanov et al., 2010*). Each DSBs is resected to form single stranded DNA, enabling homology search and repair within the context of a proteinaceous scaffold named the synaptonemal complex (*Zickler and Kleckner, 2015*).

As an illustrative example, we focus on component 5, marking Leptotene. In this component, many of the genes required for these coordinated processes have high (top 500) loadings, including *Prdm9* itself; components of the meiotic cohesin complex *Rad21l*, *Smc1b*, *Smc3*, *Stag3* and *Esco2* (*Rankin, 2015*); components of the telomere tethering complex *Terb1*, *Terb2*, *Spdya*, and *Sun1* (*Ding et al., 2007*; *Tu et al., 2017*; *Wang et al., 2019*); genes involved in creating DSBs *Mei1*, *Ccdc36* (*Iho1*), *Spo11* partner *Top6bl* (*Gm960*), and regulator *Atm* (*Lukaszewicz et al., 2018*; *Reinholdt and Schimenti, 2005*; *Robert et al., 2016*; *Stanzione et al., 2016*; *Vrielynck et al., 2016*); proteins required for the creation and processing of the ssDNA intermediates and their regulators: *Mcm8*, *Dmc1*, *Rad51*, *Rad51ap2*, *Atr*, *Brca2*, *Tex15*, *Meilb2* (*Hsf2bp*), *Meiob*, and *Spata22* (*Brown et al., 2015*; *Brown and Bishop, 2015*; *Dai et al., 2017*; *Kovalenko et al., 2006*; *Lee et al., 2015*; *Martinez et al., 2016*; *Pacheco et al., 2018*; *Ribeiro et al., 2018*; *Widger et al., 2018*; *Xu et al., 2017*; *Yang et al., 2008*; *Zhang et al., 2019*); class I crossover (ZMM group) proteins *Shoc1* (*Zip2* orthologue), *Tex11* (*Zip4* orthologue), *Msh5*, *Hfm1* (*Mer3* orthologue) and regulator *Brip1* (*FancJ*) (*Adelman and Petrini, 2008*; *Guiraldelli et al., 2018*; *Guiraldelli et al., 2013*; *Rakshambikai et al., 2013*; *Sun et al., 2016*); as well as components of the synaptonemal complex *Sycp1*, *Sycp2*, *Sycp3*, *Syce2*, *Syce3*, *Tex12*, and *Six6os1* (*4930447C04Rik*) (*Gómez-H et al., 2016*; *Syrjänen et al., 2014*). (*Figure 3D–F*; *Supplementary file 3*).

Strikingly, this component is highly enriched for GWAS hits of recombination rate in humans (*Halldorsson et al., 2019*). Of the 24 significant GWAS loci identified with confidently associated causal genes, more than half (13) rank within the top 300 genes of this component, and almost all (20) rank within the top 1300 genes (p=$5.2\times10^{-18}$, OR = 77.8 and p=$2.4\times10^{-20}$, OR = 70.1 respectively by Fisher's exact test [FET], *Figure 3F*). One hit, *Msh4*, is not ranked highly in this component (2,734[th] out of 19,262). However, MSH4 is known to function as a heterodimer with MSH5, ranking 34[th] (*Rakshambikai et al., 2013*). Unlike GWAS single cell RNA-seq does not rely on the presence of (perhaps rare, small effect) genetic variants for target discovery, while automatically identifying genes rather than SNPs affecting unknown causal genes. For example a previous GWAS (*Kong et al., 2014*) had identified a SNP in the intron of *Ccdc43*, however our expression data strongly suggested the adjacent gene *Meioc* (aka *C17orf104*) as the causal gene (ranked 183[rd] vs 13,651[st] in component 5), providing additional evidence relative to reports that *Meioc* is responsible for maintaining an extended meiotic prophase (*Abby et al., 2016*; *Soh et al., 2017*). Indeed the lead SNP in this region in a more recent GWAS is in the promoter of *Meioc* (*Halldorsson et al., 2019*). The strong enrichment of genes involved in recombination in this component suggests other highly ranked genes of unknown function could also play key roles in this process. During the

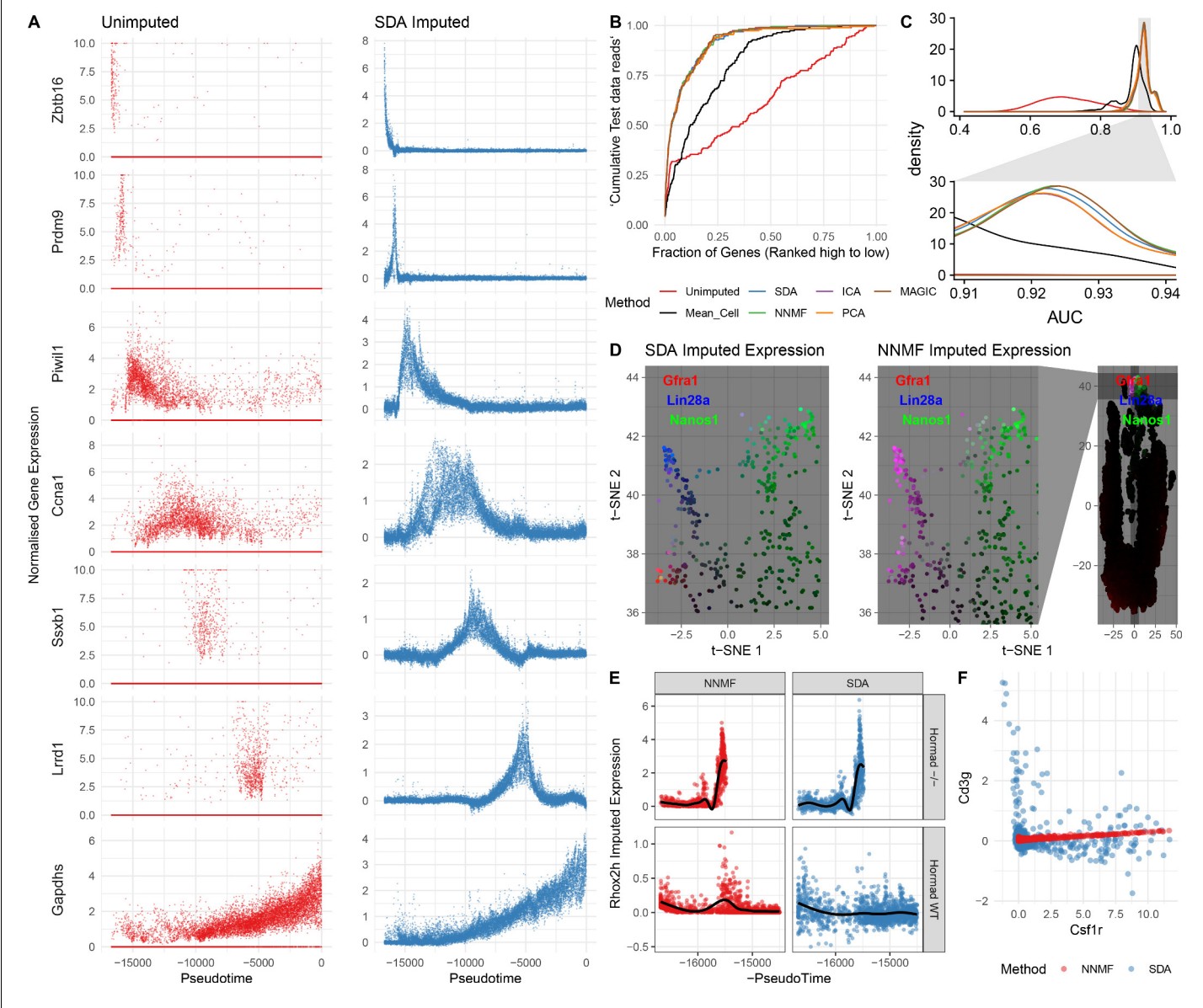

**Figure 5.** Evaluation of imputation using the SDA model. (A) Here, we illustrate the ability of SDA-based imputation (Materials and methods) of gene expression values in single cells to improve the signal/noise ratio of expression, for seven genes with strong developmental regulation. Note in the imputed expression 'dropouts' at 0 are recovered and there is less outlying expression. (B) To test the utility of SDA-based imputation, we created separate training/test data (Materials and methods). From the training data we constructed seven predictors of gene expression in the test data for each cell ('Unimputed' using the training data directly, 'Mean Cell' using the mean across all cells, matrix factorisation approaches SDA, PCA, ICA, NNMF, and a dedicated imputation approach, MAGIC). We compared the ability of each predictor to rank the gene expression in the test data for each cell, quantified as the area under the Rank Prediction Accuracy Curve (RPAC). Shown is an example RPAC for these predictors when applied to the test data for a single cell. (C) Comparison of AUCs (Area under the RPAC curve) for all cells using various methods (same color scheme as part B). (D) SDA produces multiple components for spermatogonia. Shown are zoomed in versions of the t-SNE projection (with full t-SNE for context): cells are colored by expression using a three channel ternary color scheme with the amount of blue, green, red representing the respective expression levels of *Lin28a*, *Nanos1*, and *Gfra1*. By assigning only one component for undifferentiated spermatogonia, NMF predicts *Gfra1* and *Lin28a* are expressed in the same cells resulting in a pink hue (See also *Figure 5—figure supplement 1B*, no correlation for SDA component 50 Gfra1 Stem Cells). For selection of component see Materials and methods. (E) Imputed expression of X chromosomal gene *Rhox2h* from either the SDA or NNMF decomposition, split into cells we know to be either WT or Hormad^{-/-} genotype. NNMF predicts a peak in *Rhox2h* expression even in the WT cells, in which X chromosome activation due to *Hormad1* KO does not occur. (F) NNMF does not assign separate components for the innate and adaptive immune cells (See also *Figure 5—figure supplement 1B*, no correlation for the SDA component 3 Lymphocytes). NNMF does not predict high expression of the adaptive immune cell marker *Cd3g* (T-cell surface glycoprotein CD3 gamma chain), and when it predicts any expression it increases linearly with the innate

*Figure 5 continued on next page*

*Figure 5 continued*

immune cell marker *Csf1r* (Macrophage Colony-Stimulating Factor 1 Receptor, or *Cd115*). SDA on the other hand correctly predicts that *Cd3g* and *Csf1r* are not coexpressed in the same cells.

DOI: https://doi.org/10.7554/eLife.43966.020

The following figure supplement is available for figure 5:

**Figure supplement 1.** Imputation from SDA and Other Matrix Factorization Methods.

DOI: https://doi.org/10.7554/eLife.43966.021

preparation of this manuscript, two such genes were identified: *Ankrd31* (ranked 102[nd]) plays a role in controlling the number, timing, and location of double strand breaks in meiosis (*Boekhout et al., 2019*; *Papanikos et al., 2018*), while *Hsf2bp* (now *Meilb2*, ranked 194[th]) was found to be a master regulator of meiotic recombinases (*Zhang et al., 2019*).

One striking candidate gene is *Zcwpw1*, which ranks 3[rd], after *Prdm9*. This gene does not have a known function, but contains two protein domains: CW and PWWP, known to bind H3K4me3 and H3K36me3 respectively (*He et al., 2010*; *Rona et al., 2016*). PRDM9 deposits both H3K4me3 and H3K36me3 at sites it binds (*Powers et al., 2016*), and this methyltransferase activity is essential for its role in double strand break positioning (*Diagouraga et al., 2018*), suggesting these marks may be recognized by downstream protein(s). An obvious hypothesis is that ZCWPW1 might co-localize to recombination hotspots, by binding the histone modifications deposited by PRDM9. Further work will be required to test this, and the potential role of ZCWPW1 in meiotic recombination.

The early pachytene components 13 and 47 are enriched for genes involved in the meiotic cell cycle (e.g. *Ccna1*, *Cdk1*), chromosome pairing and segregation (e.g. *Sycp3*, *Dmc1*, *Hormad1*), nuclear division (e.g. *Cenpe*, *Plk1*), and piRNA processing (e.g. *Tdrd1*, *Tdrd5*, *Tdrd9*, *Piwil1* and *Piwil2*). The next component in the temporal sequence, 48, is restricted to a small cluster of cells in t-SNE space, and enriched for many genes involved in axoneme/cilia assembly (multiple members of the *Cfap* family and dynein genes) and a smaller number of genes involved in microtubule/spindle formation (e.g. *Dcdc2b*, *Ccdc88a*, *Knl1*) and RNA splicing (e.g. *Srrm2*, *Tra2a*, *Srek1*). Components 42 and 39 (pachytene/late pachytene) are enriched for distinct genes, enriched for similar biological functions - such as meiotic cell cycle, cilium assembly, piRNA processing, and translational suppression. These two components, as well as component 47, are significantly enriched for genes that are targets of the transcription factor MYBL1 (as determined by ChIP-Seq, *Figure 7—figure supplement 1*).

The pachytene components have a striking lack of genes loading on the X or Y chromosome (*Figure 6E*), due to meiotic sex chromosome inactivation (MSCI), which is part of a broader mechanism silencing unsynapsed chromatin (MSUC) (*Turner, 2015*; *Turner, 2007*). MSCI is an evolutionarily conserved phenomenon essential for proper spermatogenesis in mammals. As previously reported (*Chen et al., 2018*; *Green et al., 2018*; *Lukassen et al., 2018*) we observe MSCI from the start of pachytene (*Figure 6A* and *Figure 4—figure supplement 3D*). Although previous bulk RNA-seq studies suggested that some genes escape MSCI (*da Cruz et al., 2016*; *Soumillon et al., 2013*), we were unable to confidently identify any genes escaping MSCI. A small number of sex-chromosome transcripts identified in pachytene cells were observed; however, these genes were highly expressed in neighboring Sertoli cells, suggesting low-level contamination as the most likely explanation. Moreover, our data indicate that previously identified 'escapees' are actually expressed after MSCI, yet fully silenced within MSCI (*Figure 6C* and *Figure 6—figure supplement 1B*).

In addition to MSCI there is the potential for lack of sex chromosome transcripts later in post-meiotic cells as they have haploid genomes possessing either an X or a Y chromosome but not both. However, cytokinesis does not fully complete in spermatogenesis resulting in synchronized chains of hundreds of cells, connected by μm-wide cytoplasmic bridges through which mRNA (or perhaps even mitochondria) could be shared (*Greenbaum et al., 2011*). The extent to which mRNA sharing occurs is unknown, but it is a property of interest to evolutionary biology as most models predict a strong fitness benefit to fathers who can mask haploid selection in their gametes (*Otto et al., 2015*). Here, we find that, with respect to sex chromosome transcription, the genetically haploid cells are predominantly phenotypically diploid (*Figure 6A & B*, and *Figure 6—figure supplement 1A*) suggesting that cytoplasmic mRNA is efficiently shared, consistent with studies of individual genes

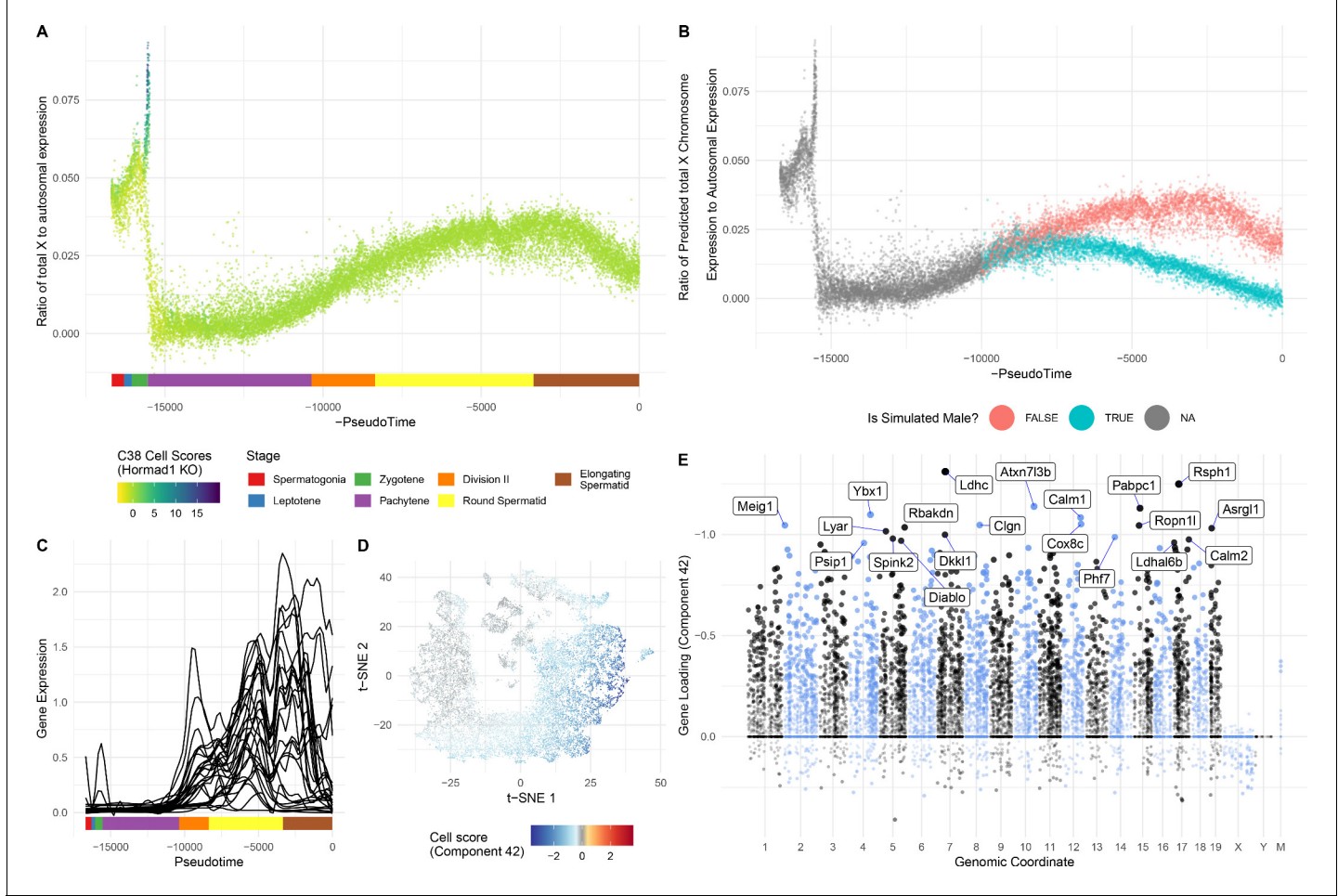

**Figure 6.** Insights into sex chromosome biology from SDA. (**A**) Pseudotime analysis provides quantitative, high-resolution insights into meiotic sex chromosome inactivation (MSCI). The sum of imputed expression for all genes on the X chromosome divided by that of the autosomes (y-axis) drops to almost 0, showing near-complete MSCI before gradually partially recovering. A similar profile is observed for genes on the Y chromosome (*Figure 6—figure supplement 1A*). (**B**) We do not observe that haploid cells obviously split into two populations due to lack of sex chromosome transcript sharing, in part A. Here were simulate what we might expect to see if there was indeed a lack of sharing (Materials and methods). (**C**) No evidence supporting prior report of genes escaping MSCI. Smoothed expression values (unimputed, gam smoothing with formula 'y ~ s(x, bs = ad)") are shown for each gene reported to escape MSCI (*da Cruz et al., 2016*) excepting *H2al1e*, *H2al1c*, and *Gm10096* which were below our dataset's expression detection threshold. Expression profiles for individual genes are separated in *Figure 6—figure supplement 1B*. (**D**) Component 42 (Pachytene) cell scores in t-SNE space. (**E**) Component 42 gene loadings. This component represents genes active during the pachytene stage of meiosis; note the striking lack of sex chromosome gene loadings, due to MSCI.

DOI: https://doi.org/10.7554/eLife.43966.022

The following figure supplement is available for figure 6:

**Figure supplement 1.** Single-gene analysis of MSCI.

DOI: https://doi.org/10.7554/eLife.43966.023

(*Braun et al., 1989*) and recent scRNA-seq reports (*Chen et al., 2018*; *Green et al., 2018*). However, there remains a possibility that some genes are not shared, such as has been observed for autosomal genes in a mutant heterozygous context: the t-complex responder mutant ($Smok^{Tcr}$) which functions as an antidote in the poison-antidote meiotic drive system of the t-complex (*Véron et al., 2009*) and Spam1 which causes transmission ratio distortion in Robertsonian (Rb) translocation-bearing mice (*Martin-DeLeon et al., 2005*).

Component 20 is particularly interesting, containing genes likely to be functional during meiotic divisions and perhaps afterwards. It contains a number of genes known to be expressed in diplotene and/or key regulators of cell division, in addition to the *Ssx* family of genes (discussed further below)

and also shows very strong enrichment of genes characterized by the presence of a DUF622 domain (18 in the top 88 genes) (*Supplementary file 3*). This rodent-specific gene family arose from duplication of the gene *Dlg5* (*Church et al., 2009*). It was previously shown that many, autosomal, DUF622 genes experience similar epigenetic changes to the sex chromosomes during spermatogenesis (*Moretti et al., 2016*). Another component (9) is most active at a similar time to 20, and is very highly enriched for genes of the electron transport chain (p=7.4×10$^{-53}$, OR = 104, FET) (*Figure 4— figure supplement 4E&F*).

We identified seven post-meiotic components characterizing wild-type biology. Round spermatid component 30 contains many genes associated with the acrosome, an organelle which forms a nuclear cap containing hydrolytic enzymes used in fertilization (*Ito and Toshimori, 2016*) (*Supplementary file 3*). For one high loading gene, *Lrrc34*, we verified by immunofluorescence that the protein is indeed localized to the acrosome of round spermatids (*Figure 2B*). Component 35, which is essentially concurrent to component 30 in pseudotime, is the most mysterious of all components that we detected. Dozens of protein-coding genes in this component are highly enriched in testis expression but have no known function (*Supplementary file 3*). This component also harbors a substantial number of genes with no apparent ortholog in humans. The existence of such a set of poorly characterized genes likely reflects the difficulty of studying postmeiotic male germ cells - which cannot be differentiated in vitro, host numerous cell-type specific processes, and express many rapidly evolving genes.

The spermiogenesis components 17, 18 and 34 all contain many genes known to be expressed at the latest stages of spermatogenesis, before transcriptional arrest due to replacement of histones with protamines (*Sassone-Corsi, 2002*) (*Supplementary file 3*). In addition, *Abhd5* (aka CGI-58), a protein previously detected in testis lipid droplets (*Wang et al., 2015*), has high loadings specifically in these late components (17 and 18) and we show by immunofluorescence that it serves as an excellent marker of the residual body (*Figure 2B*).

In addition to components for the germ cell transcriptional programs we identified components for at least five different somatic cell types: Sertoli, Leydig, Macrophages, Peritubular Myoid Cells, and T-lymphocytes. We also find a component representing an abundant somatic cell type expressing *Tcf21* but not *Acta2*, described by *Green et al. (2018)* as an unknown mesenchymal celltype, which we identify as telocytes based on coexpression of *Cd34* and *Pdgfra* (*Kuroda et al., 2004*; *Marini et al., 2018*). Some components clearly mark multiple cell types that resolve separately in t-SNE space, while others mark groups of cells that may contain cryptic heterogeneity obscured by overlapping gene expression patterns (*Figure 4—figure supplement 3F* and *Figure 3—figure supplement 1*). We were also able to infer components for batch effects such as differences in sequencing machines and different individual mice (*Figure 4—figure supplement 4A–D*).

## Validation and interrogation of SDA components by de novo inference of transcription factor binding sites and comparison to ChIP-seq data

We hypothesized that many of the SDA components represent dynamic and finely tuned transcriptional programs. If this is true, then genes within each component would be expected to have an excess of shared transcription factor binding sites within their cis-regulatory regions. We used an existing approach (*Altemose et al., 2017*; *Davies et al., 2016*) to discover de novo motifs enriched in the promoter regions of the top 250 positive and negative genes (separately) for each component (Materials and methods, *Supplementary file 5*). We compared the resulting motifs with known motifs from the HOCOMOCO database, resulting in 16 groups of matched motifs, including one group of identified de novo motifs not clearly associated with any known transcription factor, but most similar to the binding target of ATF1 (*Figure 7A*, *Figure 7—figure supplement 2*).

Although identified independently, the identified motifs include the binding targets of multiple master regulators of spermatogenesis: *Stra8* (*Kojima et al., 2019*), *Mybl1* (*Bolcun-Filas et al., 2011*), *Rfx2* (*Kistler et al., 2015*), and *Crem* (*Nantel and Sassone-Corsi, 1996*). In addition, we identified the target of *Spi1* (aka *PU.1*), a known master regulator of macrophage differentiation (*Rosa et al., 2007*), specifically in the macrophage component 11. High and specific enrichment of ChIP-seq targets of STRA8, MYBL1, RFX2 and CREM validated our interpretation - that covariation of expression of genes within many components reflects shared transcriptional regulation (*Figure 7— figure supplement 1*).

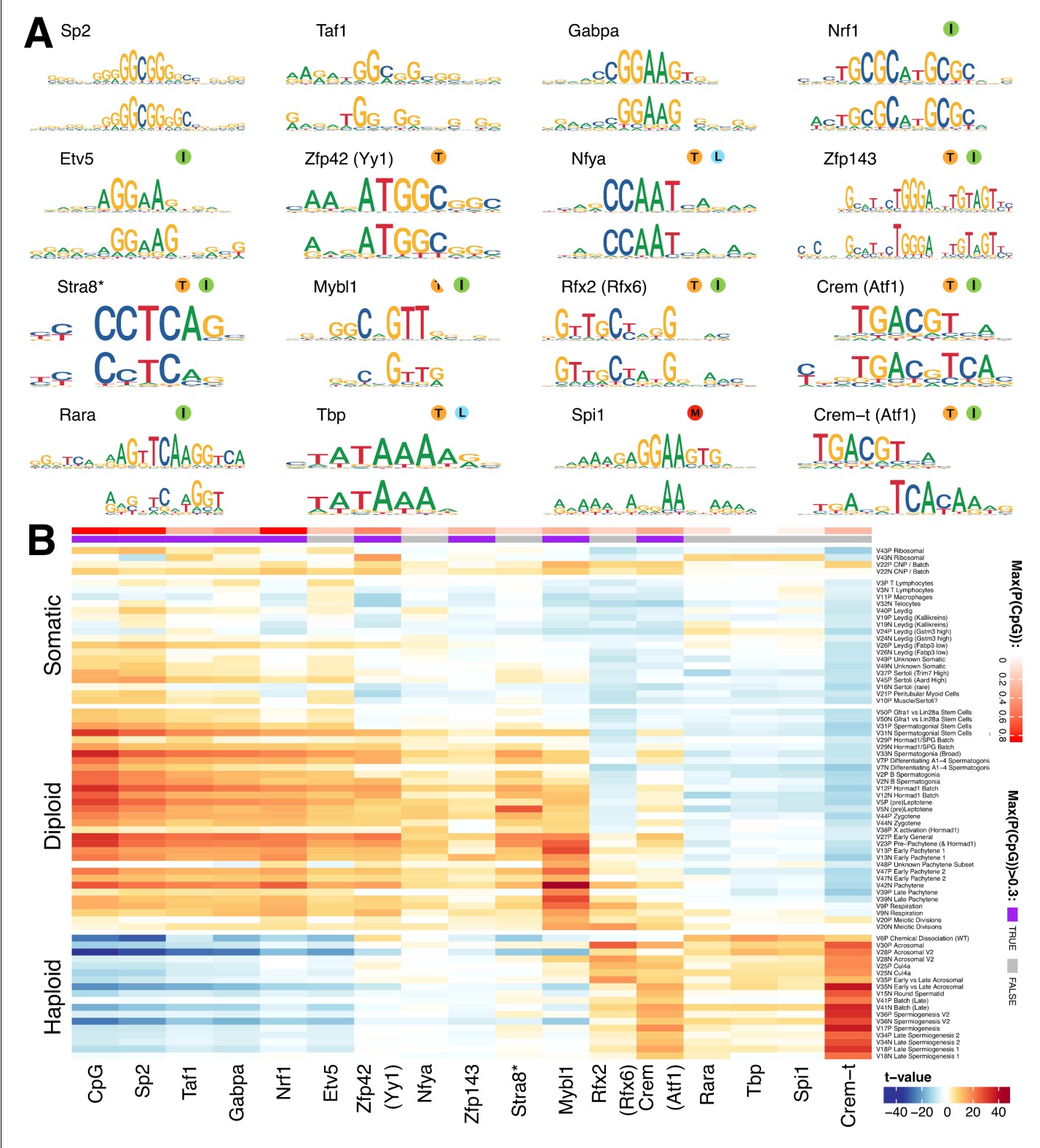

**Figure 7.** Components show shared *cis*-regulatory features. (**A**) Motifs discovered from the promoter sequences of genes with high component loadings. In each motif logo pair the lower logo shows the de novo inferred motif and the upper logo shows the motif in the HOCOMOCO database best matching the de novo motif. Orange 'T' indicates this transcription factor is highest expressed in testis in the GTEx database (half T indicates second highest). Green 'I' indicates that a mouse knockout of this gene is infertile. Blue 'L' indicates a mouse knockout of this gene is embryonic lethal. Red 'M' indicates this gene is required for macrophage development. The notation 'Crem-t (Atf1)' indicates that we suspect that the true transcription

*Figure 7 continued*

factor recognizing the motif is not the closest matching database-motif (Atf1). * the (upper) STRA8 motif shown is from Kojima et al., rather than the HOCOMOCO database (B) Association of gene loadings with the probability each de novo identified motif is found in the genes for each component. Coloring is a Z-score from a correlation test between gene loadings and motif probabilities,where red (blue) indicates positive (negative) association. The germ cell components (rows) are ordered by pseudotime. The correlation was calculated for positive and negative parts of the component separately and in the cases where the component is mainly one-sided the other side has been omitted, as have the single cell components. The additional column 'CpG' shows the same association test, but with count of promoter CpG dinucleotides, for each component. Across the top of the panel, color bars indicate the maximum probability of there being a CpG at any one position in the denovo motif, and whether that probability is greater than 0.3. See *Figure 7—figure supplement 2* for an analogous plot using the HOCOMOCO motif probabilities. We find high and specific enrichment of ChIP-seq targets of STRA8, MYBL1, RFX2 and CREM in the gene loadings of components associated with those motifs, validating our interpretation that covariation of expression of genes within many components reflects shared transcriptional regulation (*Figure 7—figure supplement 1*).

DOI: https://doi.org/10.7554/eLife.43966.024

The following figure supplements are available for figure 7:

**Figure supplement 1.** Validation of Motif Inference Using ChIP-seq Data.

DOI: https://doi.org/10.7554/eLife.43966.026

**Figure supplement 2.** Validation of Motif Inference from SDA Component Loadings.

DOI: https://doi.org/10.7554/eLife.43966.025

In our initial analysis, we frequently identified the same motif (e.g. Sp2) in multiple components. Therefore, for each motif-component combination we calculated the association between motif presence at the promoters of genes and the gene loadings (*Figure 7B*). In addition to some relatively specific enrichment (e.g. *Stra8* in Leptotene component 5, *Mybl1* in Pachytene component 42, and *Rfx2* in Acrosomal component 30), this revealed an obvious 'switch' with one group of transcription factors appearing to regulate early meiosis (prior to the meiotic division), and another group regulating meiosis post-division, with only the database Crem and Rfx2 motifs strongly spanning this divide. Moreover, most meiotic motifs spanned several components. This implies that promoter motifs might offer 'broad-scale' control, but differences at 'fine scales' among individual components might frequently also be driven by transcription factor binding to more distant enhancer regions, mRNA degradation by microRNA, or other post-translational mechanisms. Hence, additional work will be required to fully delineate the mechanisms controlling meiotic transcription.

Strikingly, many of the pre-division motifs as well as MLX/CREM contain CpG dinucleotides (*Figure 7*), with most, including the non-CpG exceptions (NFYA and ETV5) also being sensitive to DNA-methylation in their binding (*Domcke et al., 2015*; *Wang et al., 2017*). None of the post-division motifs contain CpGs, excepting one ATF like motif that we believe is instead Crem-t (see below). Indeed, we found an even stronger pattern of association simply using the count of CpG occurrences as a pseudo-motif (*Figure 7B*), indicating a major shift away from expression of genes whose promoters contain CpG islands, following the meiotic divisions. To find potential effectors of this switch we looked in the component most active at the stage of meiotic division, component 20. We found an enriched family of testis-specific proteins specifically expressed at this time, and characterized by the presence of both the SSXRD and KRAB-related domains. The SSXRD domain has been studied in the context of synovial sarcomas where it was found to associate with the CpG binding protein CXXC2 (KDM2B) which is a component of a non-canonical polycomb complex (*Banito et al., 2018*). Interestingly another non-canonical polycomb component, *Dcaf7*, also has a high loading in component 20 (*Hauri et al., 2016*). It has previously been observed that H3K27me3, a mark deposited by polycomb complex 2, increases dramatically between pachytene and the round spermatid stage (*Sin et al., 2015*). The KRAB-related domain has been studied as part of PRDM9 (the only other gene outside of the X-chromosome cluster to contain both the KRAB-related and the SSXRD domains), where it has been shown to interact with a number of proteins including the CpG binding CXXC1 (*Imai et al., 2017*; *Parvanov et al., 2017*).

As we inferred these motifs de novo we were able to discover previously unknown motifs. Indeed, we identified a motif in the late components, with partial similarity to the ATF1/CREM motifs but containing an additional CAA tail while mainly lacking the central CpG dinucleotide (*Figure 7A*). Speculatively, this may represent the binding motif of the tau isoform of CREM known to be active

in late spermatogenesis (*Sassone-Corsi, 2000*). Consistent with the more general pattern of CpG occurrence we find this ATF1/CREM-t motif highly associated with post-division components: in fact it is the most strongly associated motif in multiple such components (*Figure 7B*).

## Joint analysis of 5 mouse strains identifies pathology-related components

The flexibility of the SDA modeling framework allows the identification of sets of genes that show significant covariation in small numbers of cells. Thus, a joint analysis of mutant and wild-type cells using SDA could potentially decompose expression variation into separate technical effects, variation due to normal biological processes, and variation due to pathology, identifying mutant-specific components in the context of wild-type cells. We set out to evaluate the utility of single-cell sequencing to identify pathology in each mutant strain, combining results from both classical and SDA approaches.

Increased apoptosis is an important mechanism underlying many genetic forms of male infertility in mice. Apoptotic cells can be identified from single-cell RNA-seq data as having an excessive proportion of total transcriptome derived from mitochondrial genes (*Ilicic et al., 2016*). Cells from *Mlh3*$^{-/-}$ and *Hormad1*$^{-/-}$ animals showed higher rates of apoptosis compared to wild-type, *Cul4a* and *Cnp* (2% vs 14.5%, *Figure 1—figure supplement 1*). Pseudotime analysis provided an even finer level of resolution for staging the time of onset of developmental problems in each strain (*Figure 8A*). By performing joint pseudotime analysis on all strains simultaneously, it is in theory possible to fine map the timing of developmental defects. Our our pseudotime-ordered set of 16,950 germ cells spans the entire ~34.5 day (*Oakberg, 1957*) development process from Type A spermatogonia to mature spermatozoa, suggesting a mean difference in developmental age between pseudotime-adjacent cells of 3 minutes. Although further work is needed to clarify the mapping of pseudotime to real time, that mapping estimates the difference in the mean time of arrest of *Hormad1*$^{-/-}$ cells and *Mlh3*$^{-/-}$ cells to be 12 days. This difference is reflected in the SDA components as well; *Mlh3*$^{-/-}$ animals possess cells that load on pachytene components 47, 42 and 39, while *Hormad1*$^{-/-}$ animals do not.

HORMAD1 is a meiosis-specific protein that regulates chromosome recombination, synapsis, and segregation. HORMAD1 normally marks un-synapsed chromosomes (including sex chromosomes). While HORMAD1 is removed by TRIP13 on synapsis, it persists on asynapsed chromosomes, which then undergo MSUC, leading to MSCI for the sex chromosomes (*Shin et al., 2010*; *Wojtasz et al., 2009*). In *Hormad1*$^{-/-}$ spermatocytes, double-strand break formation and early recombination are disrupted as marked by the reduction of γH2AX, DMC1, and RAD51 foci (*Shin et al., 2010*). Hard clustering analysis (*Figure 1F & G*) showed a deficit of post-pachytene *Hormad1*$^{-/-}$ germ cells, consistent with the expectation that *Hormad1*$^{-/-}$ mutant cells experience apoptosis during meiosis I due to pachytene checkpoint failure (*Daniel et al., 2011*). Along with this arrest phenotype, the *Hormad1*$^{-/-}$ leptotene/zygotene cells form a distinct cluster outside of the leptotene/zygotene cells of all other strains (Cluster 30, *Figure 1—figure supplements 2* and *3*). A list of significant differentially expressed genes between the cluster 30 and neighboring cluster 32 included a number of sex chromosome genes (*Supplementary file 2*). Consistent with these observations, we found one SDA component (38) with much higher gene loadings on the sex chromosomes than autosomes (*Figure 9A*, *Figure 4—figure supplement 3C*, *Supplementary file 3*), and with cell loadings that are specific to *Hormad1*$^{-/-}$. We find that not only does *Hormad1*$^{-/-}$ fail to silence previously expressed sex-linked genes, but many previously *un*expressed sex-linked genes such as *Rhox2h* obtain high expression (*Figure 9B*). Interestingly, there are also multiple autosomal genes with high loadings. This may be due to ectopic expression of sex-linked transcription factors; for example, *Zfy1* and *Zfy2* were previously shown to cause pachytene arrest when misexpressed (*Royo et al., 2010*). We find a very strong association between genes in this component and genes overexpressed in mice which have mutations in either *Hormad1* or *Trip13* (p = $2.2 \times 10^{-39}$, OR = 184 and p = $1.3 \times 10^{-157}$, OR = 115, respectively by FET) (*Ortega, 2016*; *Figure 4—figure supplement 3A&B*).

CUL4A is a major component of the E3 ubiquitin ligase complex called CRL4 which is known to regulate cell cycle, DNA replication, DNA repair, and chromatin remodeling (*Dubiel et al., 2018*). Studies on *Cul4a*$^{-/-}$ mice noted that some spermatocytes arrest at the pachytene stage of meiosis I induced by the pachytene checkpoint, whereas remaining spermatocytes complete meiosis but the resulting spermatozoa present oligoasthenospermia and severe malformations (*Yin et al., 2011*).

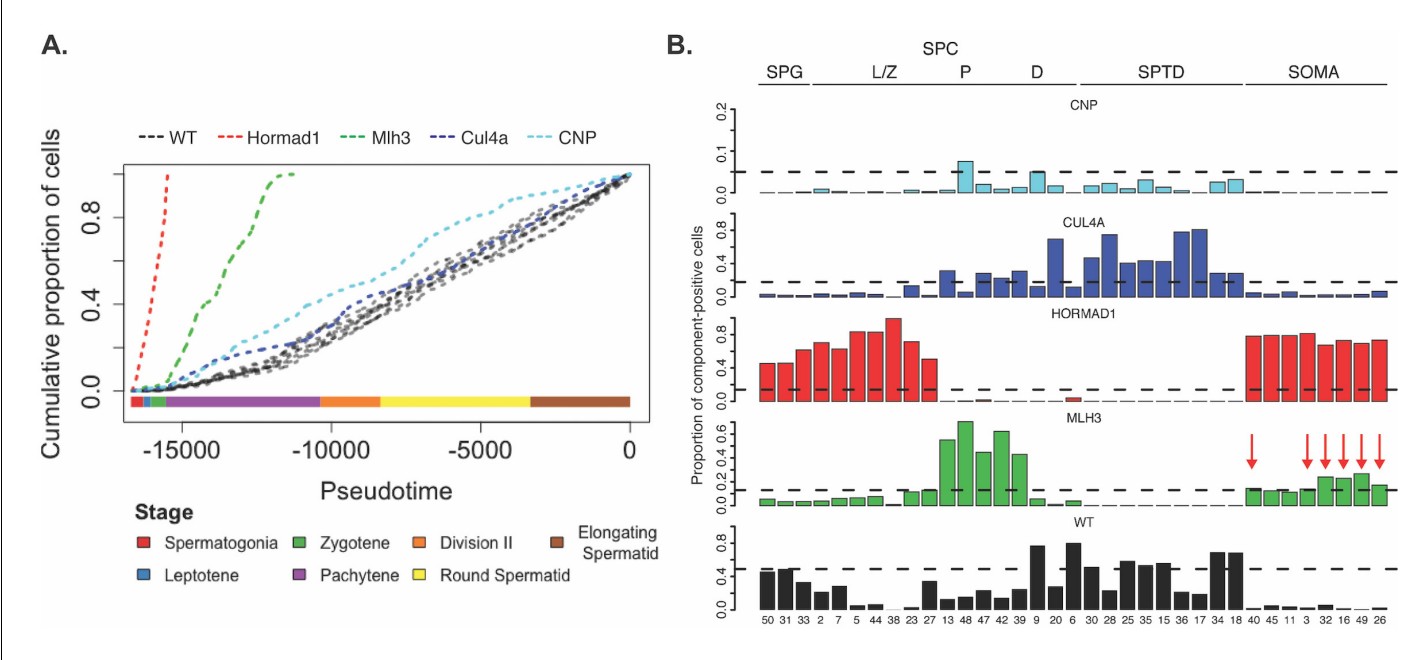

**Figure 8.** Characterization of mouse mutants with testicular phenotypes using pseudotime and SDA. (**A**) The cumulative distribution of cells along pseudotime from each mouse strain. The data clearly indicate that *Hormad1^-/-* cells arrest prior to *Mlh3^-/-* cells in the pachytene stage of spermatogenesis, while *Cul4a^-/-* and *CNP* mice show quantitative deviation from WT in the abundance of postmeiotic cells. (**B**) As a way to summarize the SDA analysis of each strain, we plot the proportion of cells with strong component loadings from each strain separately. If cells are randomly distributed across components then we would expect the fraction of cells from each mutant to be the proportion of total cells sequenced from that mutant (dashed horizontal lines). Instead there are clear enrichments of component loadings in particular mutants, providing a fingerprint of pathology for those strains. SDA components are sorted by developmental stage, as indicated by horizontal lines across the top of the panel. SPG = spermatogonial components; L/Z = leptotene/zygotene components; P = pachytene components; D = diplotene components; SPTD = components in spermiogenesis; SOMA = somatic cell components.

DOI: https://doi.org/10.7554/eLife.43966.027

The molecular basis of observed abnormal phenotypes in spermatozoa remains unclear. We identified a single SDA component (25) that was highly specific to *Cul4a^-/-* cells (**Figure 9C** and **Supplementary file 3**). This component corresponds to dozens of genes that are overexpressed in *Cul4a^-/-* mutants when compared to all other strains, with GO enrichments related to spermatid development, motility and capacitation. These findings are consistent with the observed phenotype of *Cul4a^-/-* mice and provide new leads to investigate mechanisms of pathology.

MLH3 is an essential protein required for crossover formation in early meiosis and for binding of MLH1 to meiotic chromosomes. Studies on *Mlh3^-/-* testes have shown depletion of spermatocytes and some spermatogonia due to apoptosis in diplonema induced by a reduction of chiasmata and a loss of recombination nodules (**Lipkin et al., 2002**). Interestingly, in contrast to *Hormad1^-/-*, we found no obvious transcriptional phenotype in *Mlh3^-/-* cells either by SDA analysis or by comparison of expression levels between hard-clustered wild-type and mutant cells (other than differential expression of *Mlh3*). Instead, *Mlh3^-/-* spermatocytes might simply trigger apoptosis through existing checkpoint protein machinery assembled earlier in development. Using the simple pseudotime analysis described above, we can estimate that if a transcriptional response was triggered, it might last less than approximately half an hour, for it to be missed in our sample of cells (**Figure 8A**). Similarly, the cells from *Cnp* mutant mice did not form distinct clusters, nor did they show SDA component loadings distinct from wild-type cells. Although the presence of multinucleated giant cells, hypocellular seminiferous tubules and infertile phenotype in these mice point to a serious defect in spermatogenesis, it seems difficult to determine which stages are affected using single-cell expression data. One possible explanation of missing important biological signals may be that *Cnp* mice present a partial arrest phenotype which masks the developmental abnormalities. Another possible

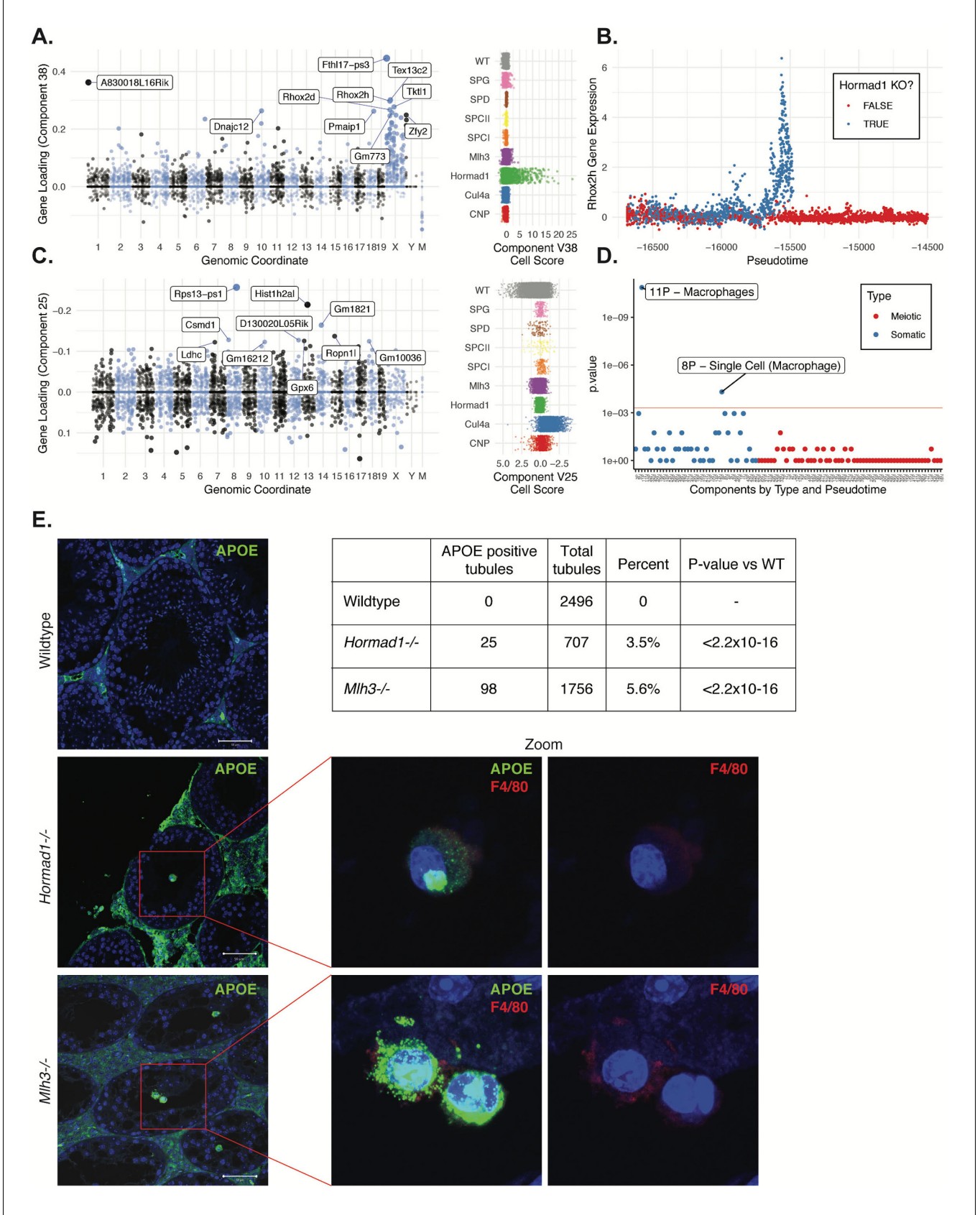

**Figure 9.** Dissection of strain-specific pathology. (A) SDA component 38 is comprised largely of genes on the X chromosome, with a gene loading direction that indicates failure of X inactivation. As illustrated by the cell scores (loadings) for this component, it is restricted to *Hormad1*[-/-] cells. (B) Pseudotime analysis indicates that *Hormad1*[-/-] cells diverge developmentally from all other strains around leptotene/zygotene. In this illustration, the X-linked gene *Rhox2h* is shown to have low or no expression in all cells prior to meiosis, and then rapidly increased expression specifically in *Hormad1*[-/-] cells until this lineage arrests. (C) Component 25 is the component most strongly enriched for *Cul4a*[-/-] cells. (D) We identified six components with shared enrichment for both *Mlh3*[-/-] and *Hormad1*[-/-] cells; these components contained genes with numerous significant GO associations related to Alzheimer's disease (AD) pathology (main text, *Figure 8B*). For each SDA component, we tested for association between known AD genes and genes with either positive (P) or negative (N) loadings on that component. AD genes are highly enriched for expression in component 11, corresponding to macrophages. (E) Further investigation of protein expression of AD genes revealed APOE+ (green) cells within the tubules of *Mlh3*[-/-] and *Hormad1*[-/-] but not WT. These cells showed nuclear morphology different from native germ cells or Sertoli cells, and stain positive for the macrophage marker F4/80. The inset table summarizes raw data on the frequency of APOE+ tubules obtained by microscopy. The frequency of APOE+ tubules is more common in each mutant strain when compared to WT by Fisher's exact test. Scale bar = 50 μm.

DOI: https://doi.org/10.7554/eLife.43966.028

explanation is that droplet-based sequencing library preparation may undersample the cells with aberrant transcriptional signatures, for example due to failure of oil droplets to encapsulate the giant cells.

## Invasion of macrophages into the seminiferous tubules is a convergent phenotype of meiotic arrest mutants

Despite the differences in cell composition or component loadings among mutant strains, we identified six somatic components (3, 16, 49, 40, 26, and 32) showing a specific enrichment for *Mlh3*[-/-] and *Hormad1*[-/-] cell loadings when compared to all other strains (*Figure 8B*). Hypothesis-free GO enrichment analysis of these components (Materials and methods) revealed a recurrence of amyloid related GO terms with qvalue <0.01, with these terms being the highest enriched term in three components (26N, 49N, 16N, *Supplementary file 6*). Excessive production of amyloid-beta, a primary cause of Alzheimer's disease, was not previously reported in these mutants, and the possible physiological role of such production is unclear. We tested multiple antibodies to human amyloid-beta that failed to work on our tissue. To further evaluate the expression of Alzheimer's disease (AD)-related genes across all five mouse strains, we tested individual SDA components for enrichment of expression of AD risk genes identified in a recent GWAS, identifying component 11 (macrophages) as specifically and strongly enriched (p=$1.3 \times 10^{-11}$, OR=65.9 by FET, *Figure 9D* and *Figure 4—figure supplement 3E*). Immunofluorescence staining for the protein product of one well-studied AD gene, *Apoe*, in wild-type animals showed low levels of specific staining, confined to the interstitial space (*Figure 9E*). Both *Mlh3*[-/-] and *Hormad1*[-/-] displayed interstitial cell with more intense staining of APOE, as well as a greater abundance of APOE+ cells. More surprisingly, we also found a rare population of APOE+ cells within the tubules of *Mlh3*[-/-] and *Hormad1*[-/-], that was never observed in wild-type. We screened 4959 tubule cross-sections to establish more precise estimates of APOE+ cell frequency in these three lines (Materials and methods). When compared to the frequency in wild-type tubules (0/2496 tubules), we see higher frequencies of intratubular APOE+ cells in *Mlh3*[-/-] (25/707 tubules, 3.5%, p<$2.2 \times 10^{-16}$) and *Hormad1*[-/-] (98/1756 tubules, 5.6%, p<$2.2 \times 10^{-16}$). These APOE+-cells displayed a nuclear staining and morphology that are distinct from normal germ cells and Sertoli cells, and appeared more similar to APOE+ cells outside of the tubules. These APOE+ intratubular cells stained for F4/80, a well-established macrophage antigen, perhaps surprisingly, given that it suggests that in these mutants, immune cells can transit the blood-testis barrier and enter an area typically regarded as immune-privileged. Intratubular macrophages have rarely been described previously, again nearly always in the context of testicular defects (*Frungieri et al., 2002*; *Goluža et al., 2014*; *Holstein, 1978*). Co-staining of F4/80 with an antibody for activated CASPASE-3, a marker of apoptosis, failed to identify any double positive cells, excluding the possibility that intratubular F4/80 protein expression was somehow an artifact of an apoptotic cell population. The mechanisms by which macrophages transit the blood-testis barrier, and the corresponding cues for migration, await further investigation.

## Discussion

The extensive cellular heterogeneity of the testis has limited the application of genome technology to the study of its gene regulation and pathology. Here, we described how the SDA analysis framework can be applied to single-cell RNA-sequencing data of the testis to overcome the challenge of heterogeneity by summarizing gene expression variation into components that reflect technical artifacts, cell types, and physiological processes. Rather than clustering groups of *cells*, SDA identifies components comprising groups of *genes* that covary in expression, and represents a single cell transcriptome as a sum of such components. This revealed previously uncharacterized complexity, with multiple different components even within recognised meiotic stages such as pachytene. This finer granularity suggests new biological interactions, for example the extremely high expression of *Zcwpw1*, a reader of specific histone modifications, within the same component as *Prdm9*, which induces identical modifications. We also identified components, both meiotic and non-meiotic, corresponding to interpretable pathology and specific to one or more mutant strains.

Other matrix factorization methods have been previously applied to soft cluster high dimensional gene expression data for example ICA, PCA (*Alter et al., 2000*; *Green et al., 2018*), Bayesian Factor Analysis (*Bernardo, 2003*) and Non-Negative Matrix Factorization (NNMF) (*Brunet et al., 2004*; *Kim and Tidor, 2003*) which naturally has a degree of sparsity in both the cell scores and gene loadings due to the positivity constraint. More recently these methods have also been applied to single cell RNAseq data (*Duren et al., 2018*; *Kotliar et al., 2018*; *Saunders et al., 2018*; *Shao and Höfer, 2017*; *Welch et al., 2019*; *Zhu et al., 2017*, reviewed in *Stein-O'Brien et al., 2018*). Here, we have reported some comparisons between SDA and these standard methods. NNMF is often motivated by the positive nature of the original data, in addition to potentially increased interpretability for purely positively additive components. However, we note that latent factors, such as those utilized by SDA, which allow negative loadings have the potential to better capture transcriptional repression. In our tests, SDA retains similar imputation performance to NNMF, while providing a more compact (in terms of sparsity) representation of the data – aiding our interpretation of components found. Beyond matrix factorization, there are other frameworks with similar goals that have been applied successfully to single cell data. One set of methods are those based on neural networks, such as self-organizing maps (*Löffler-Wirth et al., 2015*) and deep-network autoencoders (DCA) (*Eraslan et al., 2019*). DCA, much like t-SNE, creates a nonlinear embedding of the high dimensional data resulting in a lower dimensional set of scores for each cell. This approach does not, however, provide the equivalent to gene loadings and so one would have to do additional differential expression analysis on a hard clustering of the latent embeddings in order to find genes associated with the latent dimensions. To assist comparison of SDA to other methods with overlapping objectives, we have summarized resource usage of SDA across a variety of run parameters, and input data sizes (*Supplementary file 7*).

By performing de novo motif analysis, we observed that it is possible to identify transcription factors critical for the meiotic program without prior knowledge, as well as other motifs not currently well characterized. It seems very likely that our analysis of promoters is only a first step towards what is possible here via – for example – analysis of enhancers and other regulatory sequences, and we hope that future data will allow this, working towards identifying the full set of transcription factors, and their targets, used in mammalian spermatogenesis. The apparent dramatic change away from the use of factors binding CpG dinucleotides, and whose binding is often disrupted by methylation of such dinucleotides, after the first meiotic division, is one area for such further research – whether this involves *Ssx* genes, DUF622-containing genes, and/or other factors. More generally, in combination with temporal information from pseudotime analysis, it will be possible to create a model of the cascade of gene regulation, and by comparison across species, better understand the constraints on the precise timing and ordering of regulatory events.

Finally, we note that gene expression components (for example those identified by SDA) represent an attractive way to build a dictionary of pathology of the testis. The construction of new component models using a larger panel of mutants with known pathologies will accelerate the interpretation of idiopathic mutants, and, ultimately, could provide a framework for a much more advanced diagnosis of human infertility than is currently in practice.

# Materials and methods

## Key resources table

| Reagent type (species) or resource | Designation | Source or reference | Identifiers | Additional information |
|---|---|---|---|---|
| Genetic reagent (*M. musculus*) | C57BL/6J | Jackson Laboratory | Cat# 000664 | |
| Genetic reagent (*M. musculus*) | B6.129-Mlh3tm1Lpkn/J | Jackson Laboratory | Cat# 018845 | *Lipkin et al., 2002* |
| Genetic reagent (*M. musculus*) | B6;129S7-Hormad1tm1 Rajk/Mmjax | Jackson Laboratory | Cat# 41469-JAX | *Shin et al., 2010* |
| Genetic reagent (*M. musculus*) | B6;129-Cul4a$^{-/-}$ | PMID:21624359 | | Liang Ma Lab (WUSTL) |
| Genetic reagent (*M. musculus*) | C57BL/6J CNP eGFP BAC TRAP | this paper | | Joseph Dougherty Lab (WUSTL) |
| Genetic reagent (*M. musculus*) | B6;CBA-Tg(Pou5f1-EGFP)2Mnn/J | Jackson Laboratory | Cat# 004654 | |
| Antibody | Rabbit polyclonal anti-LRRC34 (G-15) | Santa Cruz Biotechnology | Cat# sc-99549, RRID:AB_2137597 | (1:100) dilution |
| Antibody | Mouse monoclonal anti-NOP132/NOL8 (D-7) | Santa Cruz Biotechnology | Cat# sc-390011 | (1:100) dilution |
| Antibody | Mouse monoclonal anti-UKHC/KIF5B (F-5) | Santa Cruz Biotechnology | Cat# sc- 133184, RRID:AB_2132389 | (1:100) dilution |
| Antibody | Mouse monoclonal anti-ABHD5 (E-1) | Santa Cruz Biotechnology | Cat# sc- 376931 | (1:100) dilution |
| Antibody | Rat monoclonal anti-F4/80 (BM8) | Santa Cruz Biotechnology | Cat# sc-52664, RRID:AB_629466 | (1:100) dilution |
| Antibody | Rabbit polyclonal anti-cleaved Caspase-3 (Asp175) | Cell Signaling Technology | Cat# 9661 | (1:400) dilution |
| Antibody | Mouse monoclonal anti-ApoE (HJ6.3) | David Holtzman Lab (WUSTL), PMID: 23129750 | | (1:1000) dilution |
| Antibody | Goat polyclonal anti-VIMENTIN (C-20) | Santa Cruz Biotechnology | Cat# sc-7557, RRID:AB_793998 | (1:100) dilution |
| Antibody | Goat polyclonal anti-ACYP1 (K-13) | Santa Cruz Biotechnology | Cat# sc-160129, RRID:AB_2242291 | (1:100) dilution |
| Antibody | Goat polyclonal anti-CCDC62 (N-15) | Santa Cruz Biotechnology | Cat# sc-240210 | (1:100) dilution |
| Antibody | Goat polyclonal anti-UNC80 (L-16) | Santa Cruz Biotechnology | Cat# sc-165859 | (1:100) dilution |
| Antibody | CF594 donkey anti-goat | Biotium | Cat# 20116, RRID:AB_10559039 | (1:300) dilution |

*Continued on next page*

*Continued*

| Reagent type (species) or resource | Designation | Source or reference | Identifiers | Additional information |
|---|---|---|---|---|
| Antibody | Alexa Fluor 488 donkey anti-mouse | Life Technologies | Cat# A-21202, RRID:AB_141607 | (1:300) dilution |
| Antibody | Alexa Fluor 488 donkey anti-rabbit | Life Technologies | Cat# A-21206, RRID:AB_2535792 | (1:300) dilution |
| Antibody | Alexa Fluor 594 donkey anti-mouse | Life Technologies | Cat# A-21203, RRID:AB_2535789 | (1:300) dilution |
| Antibody | Alexa Fluor 594 donkey anti-rabbit | Life Technologies | Cat# A-21207, RRID:AB_141637 | (1:300) dilution |
| Sequence-based reagent | Drop-seq beads | ChemGenes | Macosko201110 | PMID:26000488 |
| Sequence-based reagent | Drop-seq reagents | PMID:26000488 | | |
| Commercial assay or kit | NexteraXT | Illumina | Cat# FC-131–1024 | |
| Commercial assay or kit | Medimachine | BD Biosciences | Cat# 340588 | |
| Commercial assay or kit | 50 um Medicon | BD Biosciences | Cat# 340591 | |
| Chemical compound, drug | Hoechst 33342 | Invitrogen | Cat# H3570 | |
| Software, algorithm | Zen | other | RRID:SCR_013672 | https://www.zeiss.com/microscopy/us/products/microscope-software/zen.html; RRID:SCR_013672 |
| Software, algorithm | STAR | PMID:23104886 | https://github.com/alexdobin/STAR | *Dobin et al., 2013* |
| Software, algorithm | Drop-seq_tools | PMID:26000488 | http://mccarrolllab.com/dropseq/ | *Macosko et al., 2015* |
| Software, algorithm | Picard Tools | other | http://broadinstitute.github.io/picard/ | |
| Software, algorithm | Samtools | other | http://samtools.sourceforge.net | |
| Software, algorithm | Seurat | PMID: 29608179 | http://satijalab.org/seurat/ | *Butler et al., 2018* |
| Software, algorithm | SDA | PMID: 27479908 | https://jmarchini.org/sda/ | *Hore et al., 2016* |
| Software, algorithm | SDAtools | PMID: 27479908 | https://github.com/marchinilab/SDAtools (archived at 10.5281/zenodo.3233974) | *Hore et al., 2016* |
| Software, algorithm | TomTom | PMID: 17324271 | http://meme-suite.org/tools/tomtom | *Gupta et al., 2007* |
| Software, algorithm | MotifFinder | PMID: 29072575 | https://github.com/MyersGroup/MotifFinder (archived at http://dx.doi.org/10.5281/zenodo.3234026) | *Altemose et al., 2017* |
| Software, algorithm | ggseqlogo | PMID: 29036507 | https://omarwagih.github.io/ggseqlogo/ | *Wagih, 2017* |
| Software, algorithm | edgeR | PMID: 19910308 | https://bioconductor.org/packages/release/bioc/html/edgeR.html | *Robinson et al., 2010* |

## Mice

All animal experiments were performed in compliance with the regulations of the Animal Studies Committee at Washington University in St. Louis under protocol #20160089. Mice were housed in a barrier facility under standard housing conditions with *ad libitum* access to food and water and a 12 hr:12 hr light/dark cycle. All single-cell RNA sequencing experiments were carried out with sexually mature animals (ages of mice in this paper vary from 11 to 38 weeks) except for *Pou5f1-EGFP* transgenic animal testes which were collected at post-natal age (P) 7. For specific age of mouse at the time of testes collection for different batches, please refer to *Supplementary file 1*. Samples for histological studies were also collected at the time of testes collection for single-cell RNA sequencing. The mouse lines used in this paper are the following:

1. C57BL/6J male mice were used for Hoechst-FACS and total testis single-cell RNA sequencing experiments.
2. B6;CBA-Tg(*Pou5f1*-EGFP)2Mnn/J reporter mice were used for enriching and isolating spermatogonia type A cells. Testes from five mice at post-natal age P7 were pooled to generate single-cell suspension and FACS sorted for GFP positive cells, followed by Drop-seq.
3. B6.129-*Mlh3*$^{tm1Lpkn}$/J heterozygotes were bred to maintain the colony and male homozygotes were used for Drop-seq experiments.
4. B6;129S7-*Hormad1*$^{tm1Rajk}$/Mmjax heterozygotes were bred to maintain the colony and male homozygous knockouts were used for Drop-seq experiments.
5. B6;129 *Cul4a*$^{-/-}$ mice were used for generating Drop-seq data
6. C57BL/6J CNP-EGFP BAC-TRAP mice were used for Drop-seq data

## Single-cell suspension preparation

### Mechanical dissociation of testes

Two different types of testicular dissociation protocols were used in this paper: enzymatic and mechanical. Both enzymatic and mechanical protocols were previously published in *Getun et al. (2011)* and *Geisinger and Rodríguez-Casuriaga (2010)*. These methods were modified appropriately for single-cell RNA sequencing. For mechanical dissociation method, fresh testes were decapsulated in 1X DMEM and cut into small pieces (approximately 2–3 mm$^3$). These tissue fragments were transferred to a 50 μm Medicon, a tissue disaggregator and tissue fragments were dissociated in 1 mL 1X DMEM for 5 min on Medimachine. The resulting single-cell suspension was aspirated from Medicon with a 3 mL needless syringe. This dissociation/aspiration step was repeated three times and total of 3 mL single-cell solution was retrieved. Then the single cells were filtered through sterile 40 um strainers twice and triturated for 1 min with a wide orifice disposable Pasteur pipet. Cells were spun down at 500xg for 10 min at 4˚C and re-suspended in 2 mL 1X DMEM. Finally, cells were filtered once more with sterile 50 um filter, adjusted to 100 cells/μl concentration, and placed on ice until processed for Drop-seq. Single-cell RNA sequencing experiments were performed within ~30 min of testes collection for mechanical dissociation.

### Enzymatic dissociation of testes

Solutions necessary for enzymatic dissociation were prepared fresh prior to testes collection and these solutions are as follows: 120 U/mL collagenase type I in 1X DMEM; 50 mg/mL trypsin in 1 mM HCl; 1 mg/mL DNase I in 50% glycerol. For enzymatic dissociation method, decapsulated fresh testes were collected in 15 mL conical tubes, one testis per tube. Each testis was dissociated in 6 mL of collagenase type I solution and 10 μl of DNAse I solution with horizontal agitation at 120 rpm for 15 min at 37˚C. Tubules were decanted for 1 min vertically at room temperature and supernatant was discarded. Another 4 mL of collagenase type I solution, 50 μl of trypsin solution and 10 μl of DNAse I solution were added to each tube and incubated with horizontal agitation at 120 rpm for 15 min at 37˚C. Testicular tubules were triturated with a plastic disposable Pasteur pipet with a wide orifice for 3 min. Another 30 μl of Trypsin solution and 150 μl of DNAse I solution were added and incubated for 10 min with horizontal agitation at 120 rpm. Then 400 μl Fetal Bovine Serum (FBS) was added to deactivate dissociation enzymes. Finally, collected single-cell suspension was passed through 40 μm filter twice and stored on ice until processing for Drop-seq. These cells were processed within 1.5 hr of the testes collection.

For digesting *Pou5f1*-EGFP mice testes, we adapted a protocol described previously (*Garcia and Hofmann, 2012*). Briefly, testicular tubules/fragments were incubated in 200 µg/mL trypsin solution for 15–20 min with intermittent pipetting followed by 300 µl FBS addition for inactivating trypsin. Single-cells suspension was filtered through 50 µm filters twice and stored on ice until FACS.

## Isolation of germ cell populations by flow cytometry
### Hoechst-FACS for spermatocytes and spermatids
For isolation of major germ cell populations, we adapted a Hoechst-FACS protocol and sequential gating strategies described in *Lima et al. (2017)*. Briefly, 10 µl Hoechst and 2 µl of propidium iodide (PI) were added to single-cell suspension obtained from one testis and incubated at room temperature for 20 min. Then single-cell suspension was filtered through a 50 um cell strainer. Cells were sorted and analyzed using Beckman Coulter MoFlo Legacy cell sorter and Summit Cell sorting software. First, debris were excluded based on forward scatter (FSC) and side scatter (SSC) plot pattern. Single cells were gated by adjusting FSC and pulse width threshold. Dead cells were gated and removed based on PI intensity. A minimum of 500,000 events were observed before proceeding to gating on different germ cell populations. Then, cell count histogram was plotted based on Hoechst blue fluorescence and observed three peaks, representing haploid (1C), diploid (2C), and tetraploid (4C) populations. Then Hoechst-blue and Hoechst-red fluorescence intensities were plotted to refine spermatocytes and spermatids populations.

### Spermatogonia type A
For isolating spermatogonia type A cells from the *Pou5f1-EGFP* reporter mice, cells were analyzed and sorted with the same cell sorter and software described above section. Similar sequential gating strategies were followed. Debris were excluded, single cells were gated and dead cells were excluded. Then, GFP+ cells were gated on a plot of GFP vs FSC.

## Single-cell RNA sequencing library generation
### Drop-seq procedure
Drop-seq sequencing libraries were generated according to the protocol described previously (*Macosko et al., 2015*). Cells and beads were diluted to co-encapsulation occupancy of 0.05. Two bead lots were used for generating Drop-seq data (For more details, see *Supplementary file 1*). Individual droplets were broken by perfluorooctanol, followed by bead harvest and reverse transcription of hybridized mRNA. After Exonuclease I treatment, aliquots of 2000 beads were amplified for 14 PCR cycles (all necessary PCR reagents and conditions were identical to *Macosko et al., 2015*). PCR products were purified using 0.6x AMPure XP beads and cDNA from each experiment was quantified by Tapestation analysis. 600 pg of cDNA was tagmented by Nextera XT with the custom primers, P5_TSO_Hybrid and Nextera 70X. The single-cell sequencing library from each batch was either pooled with another batch or sequenced separately on the Illumina HiSeq2500 at 1.4pM or MiSeq at 8pM, with custom priming (Read1CustSeqB Drop-seq primer).

## Histological methods
### Collection and processing of testes
For histological studies, testes were collected in 4% paraformaldehyde (PFA), incubated overnight at 4°C and washed with 70% ethanol. For hematoxylin and eosin staining, testes were collected in modified Davidson fixative and after 24 hr incubation at room temperature, tissues were transferred to Bouin's solution for another 24 hr incubation at room temperature. Fixed testes were dehydrated through a series of graded ethanol baths and embedded in paraffin. Then 5 µm sections were cut on clean glass slides.

### Hematoxylin and Eosin (HE) Staining
Hematoxylin and Eosin staining was performed on each mouse line (Wildtype, *Mlh3*[-/-], *Hormad1*[-/-], *Cul4a*[-/-], and *CNP-EGFP*) to assess overall morphology of testicular tissue. Slides were deparaffinized with xylene and rehydrated through a series of graded ethanol bath to PBS. Standard HE staining protocol was adapted from Belinda Dana (Department of Ophthalmology, Washington University in St. Louis) and followed with Hematoxylin 560% and 1% Alcoholic Eosin Y 515.

## Immunofluorescence staining

Prior to immunofluorescence staining, antigen retrieval was performed by boiling slides in citric acid buffer for 20 min, and tissue sections were blocked in blocking solution (0.5% Triton X-100 +2% goat serum in 1X PBS) for an hour at room temperature. Primary antibodies were diluted to antibody-specific dilution (see Key Resources Table) and incubated overnight at 4°C in a humid chamber. Then, slides were incubated in secondary antibodies (1:300 dilution) at room temperature for 4 hr in a humid chamber. After the secondary antibody incubation, sections were stained with Hoechst (1:500 dilution), washed with 1X PBS and mounted with ProLong Diamond Antifade Mountant for visualization under confocal microscope.

## Computational methods

### Preprocessing of Drop-seq data

Paired-end sequencing reads were processed, filtered and aligned as described in *Macosko et al. (2015)*. The specific steps and tools for this process is further outlined in Drop-seq Computational Cookbook (http://mccarrolllab.com/wp-content/uploads/2016/03/Drop-seqAlignmentCookbookv1.2Jan2016.pdf). STAR aligner was used to map the processed reads to mouse genome (*Dobin et al., 2013*). A STAR indexed genome was generated using mm10 mouse genome and GRCm38 gene annotation (release version 76) with default setting. Following the alignment, digital gene expression (DGE) matrices were generated for each experimental batch (*Macosko et al., 2015*).

### Quality control for Drop-seq data

Combined raw DGEs were processed through a series of quality control and normalization steps. Cells with fewer than 200 UMI counts or fewer than 50 genes expressed were removed. Cells were also removed if their total UMI count or number of genes expressed was more than one standard deviation below the mean for that experiment. A t-SNE reduction of this dataset revealed an amorphous homogeneous group characterized by a low library size, high mitochondrial gene expression and often co-expressed genes from early and late meiosis suggesting poor quality and or doublet cells and so these were removed. Cells with a normalized mt-Rnr2 expression of greater than two were also removed. After these steps 20,322 cells and 28,893 genes remained.

Genes in the lower third of expression means were then removed and cells were normalized by square root transformation of total transcript counts per cell and genes were normalized to unit variance. All expression values were capped to maximum of 10. This results in a final matrix of 20,322 cells by 19,262 genes with a sparsity of 93.8% and a median UMI count of 1312 per cell.

### K-means clustering and differential expression analysis

K-means clustering was performed on the t-SNE result of SDA run (the version that removed likely components that represent batch effects and technical artifacts) using 'kmeans' in R, testing different numbers for 'k' (*Figure 1—figure supplement 2*) with maximum iterations set to 10,000. We first settled with 'k = 42 which slightly over-clustered the data (i.e. created more clusters than necessary) and then merged clusters that are transcriptionally indistinguishable. Briefly, a classification hierarchy tree that places transcriptionally similar clusters together was built using BuildClusterTree() function in Seurat(v2.3.0). To test for which clusters to be merged, the out-of-bag error (OOBE) method from a random forest classifier was used (implemented in Seurat via AssessNodes() and MergeNode() functions). The classification error was computed for left or right cells on each node of the tree and top five nodes with high OOBE were merged to finally produce 32 clusters in 'merged' t-SNE plot in *Figure 1—figure supplement 2*. Then, differentially expressed markers for all k-means clusters were identified using FindAllMarkers() function in Seurat with 'min.pct' parameter set to '0.25' where genes that are detected in a minimum fraction of 0.25 cells will be tested for differentially expressed genes. This differentially expressed genes list was used for assigning cell-types to each k-mean cluster and generating a list of potential novel cell-type specific markers by extracting top 10 differentially expressed genes for each cell-type and removing genes that were already annotated in the literature. A selected number of markers on this list was validated using immunofluorescence.

## Somatic cell population heterogeneity analysis using seurat

Seurat (v2.3.0) was used to subset, re-cluster and visualize somatic cell population data from joint wild-type and mutant dataset. After subsetting somatic cells from the original k-means result joint data (clusters 1, 2, 3, 4, 5, 8 and nine from *Figure 1—figure supplement 2A*), the percentage of mitochondrial genes was re-calculated and then a linear transformation was applied (using ScaleData () function in Seurat) while regressing out unwanted source of variations (percentage of mitochondrial genes, number of transcripts, number of genes and batch). PCA was performed on the scaled data to reduce the dimensionality of the data. A number of statistically significant principle components (PCs) for clustering purpose was determined by plotting and examining the variability explained by each PC in decreasing order (using PCElbowPlot() function in Seurat). For clustering somatic cells, we used PC = 18 as an input for K-nearest neighbor (KNN) graph-based algorithm implemented in Seurat (FindClusters()) along with resolution parameter set to '0.5.' We used t-SNE to visualize the data and clustering result. Differentially expressed genes (DEGs) were identified using Seurat's FindAllMarkers() function with 'min.pct' parameter set to '0.25' where genes that are detected in a minimum fraction of 0.25 cells are tested for DEGs. The DEGs list was used for performing gene ontology enrichment analysis to retrieve a functional profile for each somatic cluster. p-values were corrected using Benjamini-Hochberg.

## Sparse Decomposition of Arrays (SDA)

SDA v1.1 (*Hore, 2015*; *Hore et al., 2016*) was then run on the post-QC final matrix with 50 components for 10,000 iterations to confirm convergence (although in practice the results are almost identical after just 1000 iterations). The number of components was chosen such that there were typically between 1 and 5 single cell components across runs. Briefly, SDA decomposes a DGE into a number of components represented by two matrices. The columns vectors of the first matrix indicate how much a given component is active in each cell and the rows of the second matrix indicate which genes are active in a given component. SDA convergence was confirmed using the change in free energy, as well as the change in fraction of posterior inclusion probabilities (PIPs, probability that a gene loading is not equal to zero i.e. not in the spike) less than 0.5. The distribution of PIPs, cell scores, and gene loadings were also assessed. SDA was also run four further times with different seeds as well as with different number of components to verify stability of the results. Those components with a single high loading in one cell (1, 4, 14, 18, 46) were removed to visualize relationships between the components. To visualize and quantify the biological relationships among cells, t-SNE (without initial PCA step) was run on a version of the component scores matrix with likely technical artifacts and batch components removed, using a 'perplexity' parameter of 50, and 1000 iterations (Rtsne package; *Krijthe, 2015*; *der and Hinton, 2008*; *van der Maaten, 2014*). Technical components were manually identified as meeting one or both of the following criteria: two batches of the same mouse line had opposite or very different cell scores (components 6, 12, 22, 28, 29, 41) and or if the highest loading genes were all or mostly ribosomal or pseudogenes (components 9, 25, 43). To assess uncertainty in the t-SNE embedding, t-SNE was also run multiple times with different seeds (*Figure 3—figure supplement 5*). We also performed dimensionality reduction using UMAP and confirmed that it gave a pseudotime embedding consistent with t-SNE (*McInnes et al., 2018*) (*Figure 3—figure supplement 5*).

Note that SDA components have arbitrary sign and must be interpreted through the combination of gene and cell signs. Gene loadings and cell scores with concordant signs results in a positive expression contribution from a component, whereas discordant signs results in negative contribution.

To generate a pseudo-timeline we used a similar approach to that implemented in SCUBA (*Marco et al., 2014*). We iteratively fit a principal curve through the t-SNE plot with increasing degrees of freedom from 4 to 9, using the curve from the previous run as the starting point using the princurve package in R (*Hastie and Stuetzle, 1989*). Each cell was then assigned to its closest position on this curve, to define pseudotime for that cell. Somatic cells and the *Hormad1*[-/-] X-activated cells (component 38 score >3) were excluded during pseudotime construction, but the *Hormad1*[-/-] X-activated cells were given pseudotimes *post-hoc*. Somatic cells were defined by thresholding the cell scores of somatic components (if the absolute cell score of a given cell passed

any of the following component thresholds 26, 11, 3, 32, 45, 24 > 2; 37 > 1.5; 40 > 1; or mt-Rnr2 expression >3).

The temporal order of components was determined by using a weighted mean of the pseudotime values, where the weights are the cell scores of the component. In addition, only those cells with an absolute cell score of greater than two contribute to the mean. To calculate simulated haploid non-sharing (*Figure 6*) cells with pseudotime > −10000 were randomly split into two groups. The predicted X expression was calculated as Original X * PseudoTime/10000 + e, where is is a random normal error with mean 0 and s.d. of 3.

Computational analysis was performed using R (*R Development Core Team, 2018*). Gene ontology enrichment analysis was performed on the top 250 genes from each component (from each side) using the enrichGO function from the clusterProfiler R package in which p-values are calculated based on the hypergeometric distribution and corrected for testing of multiple biological process GO terms using the Benjamini-Hochberg procedure (*Yu et al., 2012*). Plots were created using the ggplot2 package and extensions ggrepel, ggforce, ggseqlogo, ggnewscale, ggrastr, RColorBrewer, viridis, and cowplot (*Campitelli, 2019*; *Garnier, 2018*; *Neuwirth, 2014*; *Pedersen, 2016*; *Petukhov, 2018*; *Wagih, 2017*; *Wickham, 2016*; *Wilke, 2018*). In addition the following R packages were used: data.table, Matrix (for sparse large matrix computations), biomaRt (for gene identifiers), shiny and shinycssloaders (for creating the interactive web application), ComplexHeatmap, bigmemory (for creating a low-memory shiny app), and MASS (for kernel density estimation) (*Bates and Maechler, 2018*; *Chang et al., 2018*; *Dowle and Srinivasan, 2019*; *Durinck et al., 2005*; *Gu et al., 2016*; *Kane et al., 2013*; *Sali, 2017*; *Venables and Ripley, 2002*).

Components were clustered by t-SNE, using either the absolute gene loadings or cell scores matrix (t-SNE perplexity = 2). Component names were then assigned based on known maker genes from the literature and cross checked for consistency against the distribution of components in t-SNE space. Components representing batch effects were identified by plotting cell scores by experimental batch and checking for biological subgroups with opposing cell scores.

We also ensured the KO cells were not unduly affecting the estimated components by separately performing an SDA analysis with only WT cells (normalized separately but with the same parameters). The same number of iterations, number of components, and random seed, were used. To account for rotations of the results we performed a procrustean rotation on the WT loadings matrix with the mixed loadings matrix as the target. Procrustes rotation was performed using the R package vegan (*Lin and Boutros, 2019*; *Oksanen et al., 2019*). We correlated the gene loadings of the Mixed WT and KO SDA analysis with the WT only analysis (after rotation) and found strong correspondence for those WT components which contained many cells (*Figure 3—figure supplements 2–4*).

## Validation of SDA imputation

Imputed gene expression values (the posterior means of the SDA model) were computed as the matrix product of the cell scores and gene loadings matrix from SDA.

In order to formally quantify the accuracy of SDA imputation, we performed a cross validation study comparing the ability of SDA imputation to correctly predict single cell gene expression data in a withheld sample. First, we randomly split the post-QC RNA-sequencing reads from the full dataset into two batches: with 20% probability a read is assigned to the test dataset, and with 80% probability it is assigned to the training dataset. Next we create seven predictors of gene expression levels for each cell, using the training dataset: 'Unimputed' uses the training data directly (scaled by the total UMI counts for each cell), 'Mean cell' uses the sum of training reads for each gene across all cells to predict ranks (i.e. every cell has the same prediction), the matrix factorization approaches SDA, ICA, PCA and NNMF were run on the normalized training data (normalized as described above) and imputed values calculated as the matrix product of cell scores and gene loadings, MAGIC values were computed using the Rmagic package.

To compare the accuracy of the three predictors for gene expression imputation, we evaluate an objective function for each predictor and each cell, which we call the 'rank prediction accuracy curve' or RPAC. The RPAC for each predictor is created by rank ordering all genes in a single cell by the predicted level of expression of those genes, from high-to-low, after reversing normalisations (*Figure 5*). For each rank (abscissa), the ordinate is the cumulative fraction of test data reads for all

genes up to that rank (i.e. all genes with higher predicted expression than the current rank). The RPAC is similar in spirit to a receiver operating characteristic (ROC) curve. The area under the curve (AUC) for each RPAC is informative about prediction accuracy; a completely random predictor is expected to produce an AUC of 0.5, while a method with some predictive utility will have an AUC >0.5. This allows us to prefer predictions with a higher AUC, although we note that (unlike for a ROC) even given perfect imputation, the maximum possible expected AUC is <1, because the test data is sparse and so shows considerable noise relative to the unknown truth.

In order to identify differences between SDA and NNMF (the most similar alternative method), for each gene we calculated imputed expression for both methods (not using single cell components from SDA), and calculated Pearson correlation between the two methods. We then looked for enrichment (by FET, p=0.05 after correction for multiple testing by Bonferroni) of the 500 least correlated genes in both SDA and NNMF components, finding seven enriched SDA components (3P, 16N, 4N, 10P, 50N, 46N, 8P) and 0 NNMF components. We show example genes from 3P and 50N in *Figure 5D and F*.

NNMF analysis was performed using the NNLM R package (*Lin and Boutros, 2019*) with 50 components, and a stop criterion of $10^{-5}$. ICA analysis was performed with the fastICA R package (*Marchini et al., 2017*) with 50 components. PCA was performed using the R package flashpcaR with 50 components and divisor and standardization set to 'none' (*Abraham and Inouye, 2014*). MAGIC was performed with default parameters using the R package Rmagic 1.5.0. (*van Dijk et al., 2018*).

## Transcription factor motif analysis

To discover de novo motifs enriched within each component we used the MotifFinder software - an iterative Gibbs sampler described in *Altemose et al. (2017)*; *Davies et al. (2016)*. We ran the Motif-Finder R package on the repeat masked promoter sequences from *Mus Musculus* GRCm38 of the top 250 positive and negative genes (separately) for each component. Sequences with greater than 10% masked bases were removed. For each component 10 different regions around the TSS were used (150, 200, 250 bp upstream and downstream of the TSS (separately), and 200, 300, 400, and 800 bp centered on the TSS). Each run was repeated nine times with different random seed motifs (each of length 6 bp). In each case MotifFinder was run for 1000 iterations or until convergence (defined as when standard deviation of the motif proportion is below 0.05).

The resulting de novo motifs were annotated with known motifs from the HOCOMOCO database (V11) using Tomtom from the MEME suite (*Bailey et al., 2015*; *Gupta et al., 2007*; *Kulakovskiy et al., 2018*). We subset the matches by taking the match with the minimum q-value for each HOCOMOCO target for those with a E-value of less than 0.001. This resulted in 123 different matches from 101 de novo motifs. These were manually grouped into 16 categories based on motif similarity. Within each group a single 'most likely acting' motif was chosen based on external suggestive evidence, including whether a knockout of the transcription factor (TF) causes infertility, if the TF is specifically expressed in testis by RNA-Seq data in GTEx (GTEx Analysis Release V7 (dbGaP Accession phs000424.v7.p2)), or if the TF is previously known to bind testis expressed genes by ChIP-Seq (*Lonsdale et al., 2013*).

To find good motifs with poor matches to currently known motifs we plotted the sum of the information content by the E value for each de novo motif. This identified a set of similar motifs with poor E values but large total information content – most of these matched best to ATF1.

In order to determine patterns of association of motifs with pseudotime, we performed correlation tests (using R's cor.test) between the gene loadings of each component (positive and negative loadings separately) and the probability of the motif being present at the promoter of these genes. To find probability of motif presence per gene promoter, we used MotifFinder but fixed the position weight matrix to either one of our de novo motifs (*Figure 7*) or a motif from the HOCOMOCO database (*Figure 7—figure supplement 2*) and ran MotifFinder for 20 iterations. We assessed convergence through the change in proportion of sequences containing the motif.

We used published ChIP-Seq data for the transcription factors *Stra8* (*Kojima et al., 2019*), *Mybl1* (*Bolcun-Filas et al., 2011*), *Crem* (*Kosir et al., 2012*), and *Rfx2* (*Kistler et al., 2015*) to validate our conclusion that some SDA components correspond to genes coregulated by these transcription factors. We performed a Fisher's exact test on the overlap of the genes suggested from the ChIP-Seq

studies for each of the three transcription factors with the top 500 genes for each SDA component (positive and negative loadings separately). For *Stra8* the genes are those described as 'STRA8-activated' from supplementary table 1 of Kojima et al. For *Mybl1* the genes are those determined as potential direct targets of MYBL1 (those that were bound in ChIP and were mis-regulated in *repro9* mice, from Table 1 of Bolcun-Filas et al.). For *Crem* the genes were those that are "found in the cross-section between DE genes from Kosir et al. and genes bound by CREM in testis from Martianov et al' (i.e. the top 50 genes from Kosir, et al. Supp Table 4). For *Rfx2* the genes are the genes from Kistler et al. that are bound by RFX2 by ChIP-seq and are statistically significantly downregulated at P30 (from their Table S1).

## Component enrichment analysis

The goal of this analysis (shown in *Figure 8B*) is to simply assess whether cell scores for each component are randomly distributed across strains. In our straightforward approach, we assess the most hypothesis-free null expectation: that cell component loadings are completely randomly distributed across cells, regardless of cell type or mouse strain. We use the full set of cells assessed by SDA. Each strain, $i$, contributed $p_i$ proportion of cells to the total SDA dataset, represented by the dashed horizontal line plotted for each strain. Under our most naive assumption, the proportion of cell from strain $i$ that load on component $j$, $f_{ij}$, should be equal to $p_i$. If $f_{ij} > p_i$ we say that the component $j$ is enriched in strain $i$.

## Apolipoprotein E (ApoE) Immunofluorescence Signal Quantification

To quantify the frequency of ApoE protein signal in wild-type and mutant animals, we counted the total number of intact testicular tubules present on slides and the number of tubules with ApoE protein signal using a confocal microscope at 20x. A Fisher's exact test was used to test the hypothesis that the frequency of ApoE-positive tubules was the same in wild-type, $Mlh3^{-/-}$ and $Hormad1^{-/-}$ strains.

## Data and software availability

Raw data and processed files for Drop-seq experiments are available under GEO accession number GSE113293.

R markdown files that enable simulating main steps of the analysis are available upon reasonable request. Custom R code used is available at www.github.com/MyersGroup/testisAtlas (*Wells, 2019*) and archived at DOI: 10.5281/zenodo.3233958.

SDA is available from https://jmarchini.org/sda/.

## Acknowledgements

We thank Abul Usmani for assistance with mouse husbandry and advice on *Pou5f1*:GFP reporter animals, Jeffrey Milbrandt and the WashU Genetics Department Single Cell Program for support, Liang Ma for providing $Cul4a^{-/-}$ mice, Joe Dougherty for providing *Cnp* mice, and Katinka Vigh-Conrad for assistance with figures. We also thank the Alvin J Siteman Cancer Center at Washington University School of Medicine and Barnes-Jewish Hospital in St. Louis, MO, for the use of the High-Speed Cell Sorter Core, which provided cell sorting service. The Siteman Cancer Center is supported in part by an NCI Cancer Center Support Grant #P30 CA91842. This work was supported by National Institutes of Health Grants R01HD078641 and R01MH101810 to DFC, and Wellcome Trust grants 098387/Z/12/Z and 212284/Z/18/Z to SM and 109109/Z/15/Z to DW. Research reported in this publication was supported by the Office of the Director, of the National Institutes of Health under Award Number P51OD011092 to the Oregon National Primate Research Center. The content is solely the responsibility of the authors and does not necessarily represent the official views of the National Institutes of Health.

## Additional information

### Funding

| Funder | Grant reference number | Author |
|---|---|---|
| Eunice Kennedy Shriver National Institute of Child Health and Human Development | R01HD078641 | Donald F Conrad |
| National Institute of Mental Health | R01MH101810 | Donald F Conrad |
| Wellcome | 098387/Z/12/Z | Simon R Myers |
| Wellcome | 109109/Z/15/Z | Daniel Wells |
| European Research Council | 617306 | Jonathan Marchini |
| Wellcome | 212284/Z/18/Z | Simon R Myers |

The funders had no role in study design, data collection and interpretation, or the decision to submit the work for publication.

### Author contributions

Min Jung, Formal analysis, Investigation, Visualization, Methodology, Writing—original draft; Daniel Wells, Conceptualization, Data curation, Software, Formal analysis, Investigation, Visualization, Methodology, Writing—original draft, Writing—review and editing; Jannette Rusch, Supervision, Investigation, Visualization; Suhaira Ahmad, Investigation; Jonathan Marchini, Conceptualization, Software, Supervision, Funding acquisition, Methodology, Writing—review and editing; Simon R Myers, Conceptualization, Formal analysis, Supervision, Funding acquisition, Methodology, Project administration, Writing—review and editing; Donald F Conrad, Conceptualization, Resources, Supervision, Funding acquisition, Investigation, Methodology, Writing—original draft, Project administration, Writing—review and editing

### Author ORCIDs

Daniel Wells https://orcid.org/0000-0002-2007-8978
Jonathan Marchini https://orcid.org/0000-0003-0610-8322
Donald F Conrad https://orcid.org/0000-0003-3828-8970

### Ethics

Animal experimentation: All animal experiments were performed in compliance with the regulations of the Animal Studies Committee at Washington University in St. Louis under approved protocol #20160089.

### Decision letter and Author response

Decision letter https://doi.org/10.7554/eLife.43966.040
Author response https://doi.org/10.7554/eLife.43966.041

## Additional files

### Supplementary files

• Supplementary file 1. Summary of all wild-type and mutant single-cell RNA-sequencing experiments.
DOI: https://doi.org/10.7554/eLife.43966.029

• Supplementary file 2. Summary of all differentially expressed genes in total joint wild-type and mutant cell clusters.
DOI: https://doi.org/10.7554/eLife.43966.030

• Supplementary file 3. Component overview. A table of key genes from 26 example components.
DOI: https://doi.org/10.7554/eLife.43966.031

- Supplementary file 4. A ZIP file containing results of the full SDA analysis reported in the manuscript, which can be loaded and explored in the R computing environment.
DOI: https://doi.org/10.7554/eLife.43966.032

- Supplementary file 5. A ZIP file containing all the de novo inferred motifs in MEME format, in addition to tables summarizing the best tomtom HOCOMOCO matches for each of these.
DOI: https://doi.org/10.7554/eLife.43966.033

- Supplementary file 6. GO Categories related to amyloid-beta metabolism show significant enrichment in components 49, 26 and 16.
DOI: https://doi.org/10.7554/eLife.43966.034

- Supplementary file 7. Summary of SDA runtime and memory usage for example datasets.
DOI: https://doi.org/10.7554/eLife.43966.035

- Transparent reporting form
DOI: https://doi.org/10.7554/eLife.43966.036

## Data availability

Raw data and processed files for Drop-seq and 10X Genomics experiments are available in GEO under accession number GSE113293.

The following dataset was generated:

| Author(s) | Year | Dataset title | Dataset URL | Database and Identifier |
|---|---|---|---|---|
| Jung M, Wells, DJ, Rusch J, Ahmad S, Marchini J, Myers S, Conrad DF | 2019 | A single-cell atlas of testis gene expression from 5 mouse strains | http://www.ncbi.nlm.nih.gov/geo/query/acc.cgi?acc=GSE113293 | NCBI Gene Expression Omnibus, GSE113293 |

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
