## [Decision Letter]

Thank you for submitting your article "Unified single-cell analysis of testis gene regulation and pathology in 5 mouse strains" for consideration by *eLife*. Your article has been reviewed by Patricia Wittkopp as the Senior Editor, a Reviewing Editor, and two reviewers. The following individuals involved in review of your submission have agreed to reveal their identity: Soeren Lukassen (Reviewer #2).

The reviewers have discussed the reviews with one another and the Reviewing Editor has drafted this decision to help you prepare a revised submission.

Summary:

In this paper, the authors perform scRNA-seq in mouse testis to delineate the cellularity of this complex organ. What distinguishes this work from other recently published scRNA-seq experiments in testis is that the application of a recently developed statistical framework method, SDA, to decompose single-cell RNA-seq expression matrices into biologically meaningful modules. Using large datasets of wild-type and mutant mouse testis scRNA-seq samples, this method allowed soft clustering, imputation of missing expression values, and gene regulatory network analysis. The standard of reporting here is high, with the authors doing a commendable job of making an interface to their data and their analytic code available to reviewers.

Essential revisions:

All reviewers agreed regarding the interest of applying an SDA-type analysis to scRNA-seq data – in particular when comparing wild-type and mutant samples – and its wider potential for the field sc-omics in general.

However, as you will see, there were also an important number of concerns, some of them major, that need to be imperatively addressed in a revised version. Notably, you are asked to make efforts for additional benchmarking, to better place your approach in the context of other methods; to provide better explanation to guide the readers of not-yet-described analyses; to provide relevant references to support statements; fix an important number of errors or omissions; and clarify the observed cellular phenotypes.

See the requests for amendments below.

*Reviewer #1:*

In this paper, the authors perform scRNA-seq in mouse testis to better understand the cellularity of this important and complex organ. What sets this work apart from several recently published scRNA-seq experiments in testis is that the authors demonstrate the power of applying a recently developed statistical framework (SDA) to scRNA-seq data. It is noteworthy that the standard of reporting here is top-notch, with the authors doing a commendable job of making an interface to their data and their analytic code available to reviewers.

This paper reads like two papers studying the same data; the first, a "classical" scRNA-seq analysis, the second, a complete, and methodologically novel reanalysis of the data using SDA. Aside from a single paragraph near the end, there are few, if any cross-references between these apparently independent studies. The latter half is an intriguing demonstration of the power of novel statistical framework to study scRNA-seq data from a complex tissue. The first half, however, is peppered with inconsistencies (outlined in specific comments below). In addition, although I understand the need to distil complex analyses into straightforward axis labels, in this first part, the authors have done so at the expense of necessary detail. For most figures, the axis labels require far more detail and/or explanation in the legend. In performing this review, far too much guesswork was required. Overall, my simplest recommendation to the authors is to greatly reduce or eliminate the "classical" scRNA-seq analyses, and instead focus on improving the latter half of this rather long paper for publication.

For both parts of the paper, my major concern stems from the authors' decision to perform a joint analysis of wild-type and mutants together. For the "classical" scRNA-seq analysis, I am not convinced that the decision to do so is justified. The contribution of mutants to patterns of clustering is not convincingly explored, indeed, in the only figure provided where this can be evaluated (currently Supplementary Figure 2B), it appears that a specific mutant phenotype may be responsible for half of the defined clusters! In the second part of the paper, the joint analysis is potentially less of a concern because SDA, in theory, should facilitate a natural delineation of cell-types that are not found in wild-type. Nonetheless, it also presents some issues that are not addressed; in particular, that the vast majority of somatic and "early" meiotic cells in their analyses appear to come from genetic mutants. I understand the reasons for this, however; (1) presenting this caveat to the reader comes very late in the paper, despite being important for multiple earlier analyses; (2) the authors never explicitly broach this topic and highlight to the reader that very few such cells are derived from wild-type animals; and (3) the authors do not convincingly demonstrate that the wild type SDA components are faithfully recapitulated in the joint analysis. These issues should be far more carefully explained and dealt with by the authors.

Finally, the authors have admirably compiled this huge dataset into a semi-digestible paper, but there are numerous instances where an understanding of not-yet-described analyses are required to explain the data (i.e. the interpretation of Figure 5 is aided by understanding Figure 6). Extensive work to linearize the manuscript and eliminate such forward-references would greatly aid legibility.

*Reviewer #2:*

In their manuscript, the authors describe a method to decompose single-cell RNA-seq expression matrices into biologically meaningful modules, and apply this method to a large dataset of wild type and mutant mouse testis scRNA-seq. Their method allows for soft clustering, imputation of missing expression values, and gene regulatory network analysis.

The authors describe an exciting analysis technique applied to a good quality dataset. The expression data will be of interest to scientists in the area of testis biology. The analysis method described is highly relevant in the entire field of single-cell omics, especially but not exclusively where differentiating tissues are the focus.

The biological interpretation of the results is largely sound and both SDA and its results are convincing. As a method-focused manuscript, additional benchmarking and comparison to other, similar tools is needed.

Major comments:

Figure 1C: The plot presents a higher transcript than gene count for many cells, which is impossible.

Figure 3D: Most of the gene loadings seem to be non-zero. This conflicts the sparsity claims made before (e.g. Figure 3B).

Subsection “Identification of gene modules using SDA”: For 2D matrices, other (NMF) implementations such as NNLM use L1 regularization (Kim and Park 2007), which should perform better at shrinking coefficients to zero than SDA, judging from Figure 3D. Furthermore, least-squares based methods should theoretically have a lower complexity than KL-based ones, so approximate resource usage values (CPU core hours and memory usage, running time on a reference machine) should be mentioned. Several papers have been published where NMF was used for the analysis of scRNA-seq data and should be discussed (Shao and Höfer, 2017; Zhu et al., 2017; Duren et al., 2018).

Subsection “Identification of gene modules using SDA”: The data presented for wild type components seems to stem from the joint analysis instead (identical component numbers, tSNE…). This should be clarified.

Figure 6F: The curve for *Cul4a* indicates an accumulation of later stages rather than a pachytene block, possibly a defect in late spermiogenesis. This contrasts previous literature (Yin et al., 2011, histology in Kupanja et al., 2011) and should be discussed.

Subsection “Joint analysis of 5 mouse strains identifies pathology-related components”: This pseudotime mapping could be seriously hampered by dissociation bias. A more finely grained analysis using shorter time periods should be compared to the one presented (for timing of tubule stages see Oakberg, 1956).

Subsection “Joint analysis of 5 mouse strains identifies pathology-related components”: The absence of giant cells could be explained by a lower rate of encapsulation. A microscopic image of an emulsion with cells stained with SiR-DNA and CellTracker or similar dyes would prove that multinucleate giant cells can be encapsulated. Furthermore, the cell suspension should be imaged before encapsulation to verify the presence of giant cells after cell straining.

General comment: In addition to matrix factorization, other techniques, such as autoencoders (DCA) and self-organizing maps (Scrat) have been used to extract features that go beyond plain PCA. In the case of autoencoders, they are also useful when applied to imputation tasks. Similarly, the motif analysis presented here is conceptually different from, but may yield similar results, as SCENIC. The performance and results of SDA in a scRNA-seq context should briefly be compared to these tools (resource usage, components/regulons on tSNE or heatmap as figure/supplementary figure).

---

## [Author Response]

Essential revisions:Reviewer #1:In this paper, the authors perform scRNA-seq in mouse testis to better understand the cellularity of this important and complex organ. What sets this work apart from several recently published scRNA-seq experiments in testis is that the authors demonstrate the power of applying a recently developed statistical framework (SDA) to scRNA-seq data. It is noteworthy that the standard of reporting here is top-notch, with the authors doing a commendable job of making an interface to their data and their analytic code available to reviewers.

We truly appreciate the reviewer’s thorough and detailed comments and criticism of our work. The reviewer clearly put a large amount of time into this and the comments have improved the quality and consistency of the manuscript.

This paper reads like two papers studying the same data; the first, a "classical" scRNA-seq analysis, the second, a complete, and methodologically novel reanalysis of the data using SDA. Aside from a single paragraph near the end, there are few, if any cross-references between these apparently independent studies. The latter half is an intriguing demonstration of the power of novel statistical framework to study scRNA-seq data from a complex tissue. The first half, however, is peppered with inconsistencies (outlined in specific comments below). In addition, although I understand the need to distil complex analyses into straightforward axis labels, in this first part, the authors have done so at the expense of necessary detail. For most figures, the axis labels require far more detail and/or explanation in the legend. In performing this review, far too much guesswork was required. Overall, my simplest recommendation to the authors is to greatly reduce or eliminate the "classical" scRNA-seq analyses, and instead focus on improving the latter half of this rather long paper for publication.

We understand the reviewer’s perspective, and have restructured the paper as suggested. The structure of the paper reflected the state of the field when we began putting this work together 1.5 years ago; at that point there were no “classical” scRNA-seq papers on the testis. We agree that best way to present our study now is to greatly reduce the “classical” section.

For both parts of the paper, my major concern stems from the authors' decision to perform a joint analysis of wild-type and mutants together. For the "classical" scRNA-seq analysis, I am not convinced that the decision to do so is justified. The contribution of mutants to patterns of clustering is not convincingly explored, indeed, in the only figure provided where this can be evaluated (currently Supplementary Figure 2B), it appears that a specific mutant phenotype may be responsible for half of the defined clusters! In the second part of the paper, the joint analysis is potentially less of a concern because SDA, in theory, should facilitate a natural delineation of cell-types that are not found in wild-type. Nonetheless, it also presents some issues that are not addressed; in particular, that the vast majority of somatic and "early" meiotic cells in their analyses appear to come from genetic mutants. I understand the reasons for this, however; (1) presenting this caveat to the reader comes very late in the paper, despite being important for multiple earlier analyses; (2) the authors never explicitly broach this topic and highlight to the reader that very few such cells are derived from wild-type animals; and (3) the authors do not convincingly demonstrate that the wild type SDA components are faithfully recapitulated in the joint analysis. These issues should be far more carefully explained and dealt with by the authors.

We have now introduced this important issue in the first section of the Results section, and devoted several paragraphs to quantifying the effects of our joint analysis. We have generated a new supplemental figure (Figure 1—figure supplement 4) that tabulates the count of cells from each strain within each cluster, as well as the number of genes identified as differentially expressed between mutant and wildtype within each cluster. Reassuringly, only 2/32 clusters do not contain both mutant and wildtype cells; one is a leptotene spermatocyte cluster corresponding to *Hormad1-/-* cells that fail X inactivation, and the other is a Sertoli cell cluster. Both of these lack wildtype cells. The reviewer is correct in pointing out that our somatic cells are mainly derived from mutant strains (95%) and this is now noted clearly. Due in part to our wildtype FACS experiments, we have a more balanced ascertainment of premeiotic germ cells, with 17% of spermatogonia or pre-pachytene spermatocytes derived from wildtype.

We have also introduced a new SDA analysis where we used only wild-type cells, and compared the resulting components to those in the joint analysis (see Figure 3—figure supplement 2 for results).The most important WT-only components have high correlations with components in the mixed SDA run, although mutant specific components do not appear, and some early meiotic components do not appear as these cells are enriched (only) in our mutant samples due to the lack of later cells in these samples.

Finally, the authors have admirably compiled this huge dataset into a semi-digestible paper, but there are numerous instances where an understanding of not-yet-described analyses are required to explain the data (i.e. the interpretation of Figure 5 is aided by understanding Figure 6). Extensive work to linearize the manuscript and eliminate such forward-references would greatly aid legibility.Reviewer #3:[…]Major comments:Figure 1C: The plot presents a higher transcript than gene count for many cells, which is impossible.

The labels for the subpanels were erroneously swapped, and this has now been corrected.

Figure 3D: Most of the gene loadings seem to be non-zero. This conflicts the sparsity claims made before (e.g. Figure 3B).

Unfortunately, the low resolution of this plot made it appear that the majority of the loadings were non-zero. Moreover, gene loadings with a value of 0 are plotted on top of one another which in general makes it difficult to judge sparsity from this visualisation. We have updated Figure 3D (right hand side below) to make this clearer by using varying opacity and variable point sizes. In no components are there more genes with PIP>0.5 than <0.5. The average fraction of genes with PIP<0.5 is 80%, i.e. most components are sparse.

Subsection “Identification of gene modules using SDA”: For 2D matrices, other (NMF) implementations such as NNLM use L1 regularization (Kim and Park 2007), which should perform better at shrinking coefficients to zero than SDA, judging from Figure 3D. Furthermore, least-squares based methods should theoretically have a lower complexity than KL-based ones, so approximate resource usage values (CPU core hours and memory usage, running time on a reference machine) should be mentioned. Several papers have been published where NMF was used for the analysis of scRNA-seq data and should be discussed (Shao and Höfer, 2017; Zhu et al., 2017; Duren et al., 2018).

Thanks. We have added a new paragraph about alternative matrix factorisations in the discussion, referencing the suggested recent studies applying these approaches to scRNA-seq data. We have also performed more analysis comparing SDA to other matrix factorisations such as NMF, PCA and ICA. Figure 3C now compares sparsity of these approaches. Another natural way of comparison is predictive ability of the model in the cross-validation approach we previously used to assess imputation for SDA. Figure 5B and C now compare this predictive ability for NMF, PCA and ICA in addition to SDA and other non-matrix factorisation imputation approaches including MAGIC. As NNMF performs similarly both in terms of imputation/prediction and sparsity compared to SDA we further investigated the genes which disagree the most between these two methods (lowest correlation between predicted expression). We found a number of genes/components represented better by SDA than NMF for the same number of components, which are now shown in Figure 5D-F and Figure 5—figure supplement 1A-B. We have also quantified resource usage which can now be found in Supplementary File 7.

Subsection “Identification of gene modules using SDA”: The data presented for wild type components seems to stem from the joint analysis instead (identical component numbers, tSNE…). This should be clarified.

This has been made much more explicit in the revised manuscript.

Figure 6F: The curve for Cul4a indicates an accumulation of later stages rather than a pachytene block, possibly a defect in late spermiogenesis. This contrasts previous literature (Yin et al., 2011, histology in Kupanja et al., 2011) and should be discussed.

We have remade this figure using an updated version of our pseudotime analysis. In this new version we now see an uptick of accumulation during pachytene (compared to WT) that is consistent with the pachytene block observed in the prior studies of *Cul4a*-/- animals; however, the phenotype is mild. This is also consistent with what what we observed in the histology for several of these animals. We believe the *Cul4a*-/- phenotype becomes progressively worse with age and so we might see more severe defects in older animals.

Subsection “Joint analysis of 5 mouse strains identifies pathology-related components”: This pseudotime mapping could be seriously hampered by dissociation bias. A more finely grained analysis using shorter time periods should be compared to the one presented (for timing of tubule stages see Oakberg, 1956).

We believe the reviewer is suggesting that the distribution of cells across pseudotime may not reflect the true distribution due to dissociation bias. We acknowledge that there is almost certainly some dissociation bias, but this dissociation bias should affect all strains. We believe it is valid to make comparisons among cells obtained with the same dissociation protocol. We also provide a caveat in the text: “Further work is needed to clarify the mapping of pseudotime to real time”.

We have added 7 inferred phases of spermatogenesis to the pseudotime figure (now Figure 8) to improve interpretation.

Subsection “Joint analysis of 5 mouse strains identifies pathology-related components”: The absence of giant cells could be explained by a lower rate of encapsulation. A microscopic image of an emulsion with cells stained with SiR-DNA and CellTracker or similar dyes would prove that multinucleate giant cells can be encapsulated. Furthermore, the cell suspension should be imaged before encapsulation to verify the presence of giant cells after cell straining.

We agree with the reviewer’s suggested explanation, and tried to say that with the sentence “Another possible explanation is that droplet-based sequencing library preparation may undersample the cells with aberrant transcriptional signatures”. We have rephrased this sentence for clarification, to read:

“Another possible explanation is that droplet-based sequencing library preparation may undersample the cells with aberrant transcriptional signatures, e.g. due to failure of oil droplets to encapsulate the giant cells.”

General comment: In addition to matrix factorization, other techniques, such as autoencoders (DCA) and self-organizing maps (Scrat) have been used to extract features that go beyond plain PCA. In the case of autoencoders, they are also useful when applied to imputation tasks. Similarly, the motif analysis presented here is conceptually different from, but may yield similar results, as SCENIC. The performance and results of SDA in a scRNA-seq context should briefly be compared to these tools (resource usage, components/regulons on tSNE or heatmap as figure/supplementary figure).

As the reviewer points out, matrix factorization is just one possible framework for obtaining low-dimensional summaries of high-dimensional data. Autoencoders and self-organizing maps are both methods based on neural networks, and have been applied successfully to single-cell data. We have included references to both methods in the discussion. One important distinction between DCA and SDA is that SDA is more interpretable. DCA, much like tSNE, creates a nonlinear embedding of the high dimensional data resulting in a lower dimensional set of scores for each cell. It does not, however, provide the equivalent to gene loadings and so one would have to do additional differential expression analysis on a hard clustering of the latent embeddings in order to find genes associated with the latent dimensions. It would therefore, for example, not have been possible to do the transcription factor analysis directly as we did with SDA. We have added a paragraph reviewing these considerations in the Discussion section.

To assist comparison of SDA to other methods with similar objectives, we have summarized resource usage of SDA across a variety of run parameters, and input data sizes. This is now reported as a table in Supplementary File 7.

The computational complexity of SDA scales with NLC^2^ where N=number of cells, L=number of genes, and C=number of components. We acknowledge SDA is not the fastest way to analyse single cell RNA-seq data, however we prefer additional insight over gains in speed which for datasets of similar size to ours will be small relative to the time for data generation and analysis overall. We also note that SDA can be run in parallel, and that we find almost identical results running for 1,000 iterations but used 10,000 iterations for the publication results in an abundance of caution.